



# CAMELS-INDIA: hydrometeorological time series and catchment attributes for 472 catchments in Peninsular India

Nikunj K. Mangukiya*[1], Kanneganti Bhargav Kumar*[2], Pankaj Dey[1], Shailza Sharma[2], Vijaykumar Bejagam[1], P. P. Mujumdar[2,3], Ashutosh Sharma[1,4]

* Equal contribution
[1] Department of Hydrology, Indian Institute of Technology Roorkee, Roorkee, 247667, Uttarakhand, India
[2] Department of Civil Engineering, Indian Institute of Science, Bangalore, 560012, Karnataka, India
[3] Interdisciplinary Centre for Water Research, Indian Institute of Science, Bangalore, 560012, Karnataka, India
[4] International Centre of Excellence for Dams, Indian Institute of Technology Roorkee, Roorkee, 247667, Uttarakhand, India

*Correspondence to*: Ashutosh Sharma (ashutosh.sharma@hy.iitr.ac.in)

**Abstract.** We introduce CAMELS-INDIA (Catchment Attributes and MEteorology for Large-sample Studies – India), the hydrometeorological time series, and catchment attributes for 472 catchments in Peninsular India. Peninsular India covers 15 intrastate river basins defined by the Central Water Commission (CWC), where river flow and water level datasets are available for several gauge stations through the open-source India Water Resources Information System (India-WRIS). However, many

of these gauge stations lack reliable metadata, and data are not in an analysis-ready format for large-sample hydrological studies. Therefore, we utilized 472 gauge stations and their catchment boundaries, characterized as stations with reliable metadata, from the 'Geospatial dataset for Hydrologic analyses in India (GHI)' (Goteti, 2023). For each of these catchments, the CAMELS-INDIA provides a catchment mean time series of meteorological forcings for 41 years (1980-2020) and around 211 catchment attributes representing hydroclimatic and land cover characteristics extracted from multiple data sources

(including ground-based observations, remote sensing-based products, and reanalyses datasets). The CAMELS-INDIA follows the same standards of the previously developed CAMELS datasets for the USA, Chile, Brazil, Great Britain, Australia, Switzerland, Germany, and Denmark to facilitate comparisons with catchments of those countries and inclusion in global hydrological studies. Notably, the CAMELS-INDIA includes available observed streamflow and catchment mean time series of 19 meteorological forcings, including precipitation, maximum, minimum, and average temperature, long-wave and short-

wave radiation flux, U and V-components of wind, relative humidity, evaporation rates from canopy and soil surface, actual and potential evapotranspiration, and soil moisture of four layers (covering depth up to 3 m below ground) for detailed hydrometeorological studies. We also derived catchment attributes representing human influences, including the number of dams and their utilization, total volume contents of dams in catchments, population density, and increase in urban and agricultural land covers to facilitate studies to understand human influences on catchment hydrology. Furthermore, the dataset

includes predicted streamflow time series from a regionally trained Long-Short Term Memory (LSTM)-based hydrological model, which can fill gaps in observed streamflow data or serve as a benchmark for testing and developing new hydrological models. We envision that CAMELS-INDIA will provide a strong foundation for a community-led effort toward gaining new



hydrological insights from hydrologically distinct Indian catchments and solving pertinent issues related to water management, quantification and risk assessment of hydrologic extremes, unraveling regional-scale hydrologic functioning, and climate
change impact assessment of catchments across India. The CAMELS-INDIA dataset is available at https://doi.org/10.5281/zenodo.13221214 (Mangukiya et al., 2024).

## 1 Introduction

Large-scale hydrological studies to formulate generalized conclusions on hydrological models and processes require data from large samples of catchments to understand spatiotemporal hydrological differences across scales (Addor et al., 2017; Coxon
et al., 2020). Various studies have utilized large-sample datasets to investigate the impacts of climate change and anthropogenic influences on hydrological behavior (Van Loon et al., 2022; Feng et al., 2023), for predictions of hydrometeorological variables (Feng et al., 2020; Kratzert et al., 2018; Lees et al., 2021; Mangukiya et al., 2023), for hydrological classification and similarities (Fang et al., 2022; Dimitriadis et al., 2021; Jehn et al., 2020), for predictions in the ungauged and data-sparse region (Kratzert et al., 2019; Ma et al., 2021; Nearing et al., 2024), and for understanding drivers of extreme events and future
hydrological changes (Mangukiya and Sharma, 2024; Alvarez-Garreton et al., 2021; Zhang et al., 2022; Das et al., 2024). The primary data required for hydrometeorological analyses are streamflow and its drivers, such as precipitation, temperature, solar radiation, evapotranspiration, wind, soil moisture, and relative humidity. Ideally, the hydrometeorological time series datasets are complimented by catchment attributes, which are believed to control hydrological processes, such as topography, land cover, soil, and geology (Addor et al., 2017). The availability of such catchment data sets provides a new perspective to the
research community for finding answers to some relevant questions that could not be addressed in the past. In addition, it helps the researchers to expedite their research by saving hours of collecting and processing the data from various sources.

The compilation of hydrometeorological time series and complimentary attributes for large samples of catchments began in 2006 with the 'Model Parameter Estimation Experiment (MOPEX)' dataset (Schaake et al., 2006) in the USA. Later, the MOPEX dataset was extended by Newman et al. (2015) and Addor et al. (2017), resulting in the first 'Catchment Attributes
and MEteorology for Large-sample Studies (CAMELS)' dataset comprising 671 catchments in the contiguous United States (CONUS). Given the importance of such a large-sample dataset for hydrometeorological studies, the CAMELS and other datasets are developed for various countries, such as Chile (CAMELS-CL; Alvarez-Garreton et al., 2018), North America (HYSETS; Arsenault et al., 2020), Brazil (CAMELS-BR; Chagas et al., 2020, and CABra; Almagro et al., 2021), Great Britain (CAMELS-GB; Coxon et al., 2020), China (CCAM; Hao et al., 2021), Australia (CAMELS-AUS; Fowler et al., 2021), Austria
(LamaH-CE; Klingler et al., 2021), France (CAMELS-FR; Andréassian et al., 2021), Switzerland (CAMELS-CH; Höge et al., 2023), Spain (CAMELS-ES; Casado Rodríguez, 2023), Sweden (CAMELS-SE; Teutschbein, 2024), Germany (CAMELS-DE; Loritz et al., 2024), and Denmark (CAMELS-DK; Liu et al., 2024). Recently, the initiative to combine all existing CAMELS and other large-sample datasets was taken through Caravan (Kratzert et al., 2023) to facilitate global hydrological studies. The cloud-based platform of Caravan for extraction of meteorological forcings and catchment attributes further





extended Caravan datasets for Denmark (Koch, 2022) and Israel (Efrat, 2023). Despite the increasing availability of large-sample hydrometeorological datasets globally, India still lacks a comprehensive dataset for large-sample hydrological studies. In India, accessing analysis-ready datasets is difficult, and the available open-source datasets require additional quality checks (Goteti, 2023). The Central Water Commission (CWC) and various state government agencies provide water-related data through the online portal, India – Water Resources Information System (India-WRIS; https://indiawris.gov.in/wris/#/).

However, the related Geographic Information System (GIS) metadata, such as digitized gauge locations, catchment boundaries, and river network information, is still limited, and researchers need to put significant efforts into digitizing and compiling the required information from available CWC reports (Goteti, 2023). For meteorological time series datasets, the India Meteorological Department (IMD) provides a nationwide gridded dataset of rainfall and temperature, and the National Centre for Medium Range Weather Forecasting (NCMRWF) provides various other meteorological variables in gridded format

through the Indian Monsoon Data Assimilation and Analysis (IMDAA) – reanalysis data services (https://rds.ncmrwf.gov.in/home). However, such nationwide datasets are rarely aggregated to the catchment scale and require pre-processing to make them analyses ready (Hao et al., 2021). For large-scale hydrological studies, searching for appropriate data, finding methods for data pre-processing, and formatting data consume considerable time and redundant efforts with limited research advances (Beniston et al., 2012; Hao et al., 2021). Due to a lack of analysis-ready datasets and associated

difficulties in data processing, unsurprisingly, large-sample hydrological studies are less common in India than in the USA or Europe. To overcome all these difficulties, community-led efforts are required to develop the needed analysis-ready dataset for India (Goteti, 2023).

Goteti (2023) recently provided the first quality controlled publicly available hydrographic dataset, the 'Geospatial dataset for Hydrologic analyses in India (GHI)', which includes GIS data on locations of gauges, catchment boundaries, and river network,

and monthly and annual time series of precipitation, evapotranspiration, and runoff for 472 catchments in peninsular India. Even though the GHI dataset does not systematically provide catchment attributes representing hydroclimatic, land cover, and anthropogenic influences, it paved the way for the hydrologic community by providing reliable GIS metadata for a consistent set of catchments for Indian river basins. To address the data gap of GHI, we produced the CAMELS-INDIA dataset (Mangukiya et al., 2024), which provides a daily catchment mean time series of 19 meteorological forcing, available observed

and Long-Short Term Memory (LSTM)-based hydrological model predicted streamflow (Mangukiya et al., 2023), and around 211 catchment attributes representing topographic, climatic, hydrologic, land cover, soil, geological, and anthropogenic influence characteristics for 472 catchments in Peninsular India. The proposed dataset will be the stepping stone to provide large-sample meteorological time series and attributes of the Indian catchments to the global and national hydrological community. The CAMELS-INDIA follows the same standards as the previously developed CAMELS datasets to facilitate

comparisons with catchments of those countries and inclusion in global hydrological studies. The following sections describe the objectives behind the CAMELS-INDIA dataset and comprehensively describe all data supplied within CAMELS-INDIA, including its data source and how the hydrometeorological time series and static catchment attributes were prepared.



## 2 Motivation and Rationale

India has hydrologically distinct catchments spread across arid, temperate, and tropical climate zones (**Fig. 1a**). These
catchments are heterogeneous in terms of characterization and are influenced to varying degrees by anthropogenic activities
(Mangukiya and Sharma, 2024; Mangukiya et al., 2023). Despite these unique characteristics, Indian catchments are often
underutilized in global hydrological studies due to insufficient analysis-ready datasets. The publication of CAMELS-INDIA
aims to address this gap, providing an essential resource for researchers worldwide to investigate hydrological regimes under
anthropogenic influences and changing climates, thus tackling water-related issues. CAMELS-INDIA includes over 100 arid
catchments, which can be combined with other arid-zone catchments, such as those in CAMELS, CAMELS-CL, and
CAMELS-AUS, enabling large-sample studies of arid-zone hydrology (Fowler et al., 2021). Furthermore, India's catchments
are regulated by large and medium dams due to the seasonality of rainfall, often experiencing water limitations on a seasonal
basis. This characteristic offers a significant number of samples to the global research community, aiding in addressing various
modeling challenges specific to catchments with such unique features.

Given the global use case, during the development of CAMELS-INDIA, a critical choice was whether to utilize national or
global datasets for extracting hydrometeorological time series and catchment attributes. While global datasets would facilitate
intercontinental comparisons, national datasets would provide the highest-quality information available in India. So far,
CAMELS datasets of different countries have utilized the best possible national data sources, drawing on the expertise of
CAMELS creators. In cases where national datasets were unavailable, global datasets, such as the 'Global Lithological Map
(GLiM)' (Hartmann and Moosdorf, 2012) and 'GLobal HYdrogeology MaPS (GLHYMPS)' (Gleeson et al., 2014), were used.
Using national products would facilitate global users, potentially unfamiliar with such products, to benefit from these local
insights (Fowler et al., 2021). It will also encourage national-scale studies by providing analysis-ready datasets from the best
available data source within the country. Moreover, ongoing efforts, such as Caravan (Kratzert et al., 2023), to produce
consistent global datasets (using global data products for deriving meteorological time series and catchment attributes) will
complement the data produced from national sources and facilitate comparative studies. Therefore, we prioritized national data
products, where possible, to produce CAMELS-INDIA.

## 3 Catchments and Data Availability

The CWC and other state government agencies have listed 4824 gauge locations, at present, on India-WRIS for users to obtain
streamflow observations. However, out of those, only 645 gauges in Peninsular India offer free access to data for users. The
remaining stations either lack data or fall under the 'classified data' category due to transboundary river basins. Given the
existing challenges in validating and extracting information from these available datasets in India, the GHI has introduced the
first quality-controlled metadata in GIS format and listed 472 catchments with reliable metadata out of the 645 gauge stations
in Peninsular India (Goteti, 2023). In the CAMELS-INDIA dataset, we have incorporated these 472 catchments located in





Peninsular India (**Fig. 1b**) to extract daily meteorological time series and catchment attributes for large sample hydrological

studies.

Peninsular India is a large region situated between the Western Ghats and the Eastern Ghats, extending south of the Vindhya range (**Fig. 1b**). The elevation ranges from 0 to 2600 m above mean sea level, with a mean elevation of 600 m, sloping eastward. The Western Ghats, also known as Sahyadri hills, are a prominent landform in this region and play a crucial role in controlling moisture movement during the southwest monsoon. The Palghat gap, a narrow region in the Western Ghats, is a

geological shear zone representing a weak area in the earth's crust. This gap supports a network of brooks and creeks forming the west-flowing Bharathappuzha river, the second-largest river in Kerala. This gap influences the weather patterns in Peninsular India by allowing moisture-laden southwest monsoon winds to enter the state of Tamil Nadu, moderating the summer temperatures and increasing the rainfall in the region. Other major landforms in Peninsular India include the Eastern Ghats, a discontinuous mountain range along the Bay of Bengal coast. These mountains are eroded and intersected by major

rivers of Peninsular India, the Mahanadi, Godavari, Krishna, and Cauvery. These rivers create large delta regions east of Eastern Ghats, with nutrient-rich soils (**Fig. 1b**). The Maikal range in the north is the origin of the Narmada River. **Figure 1b** illustrates the major river basins and gauge locations in Peninsular India.

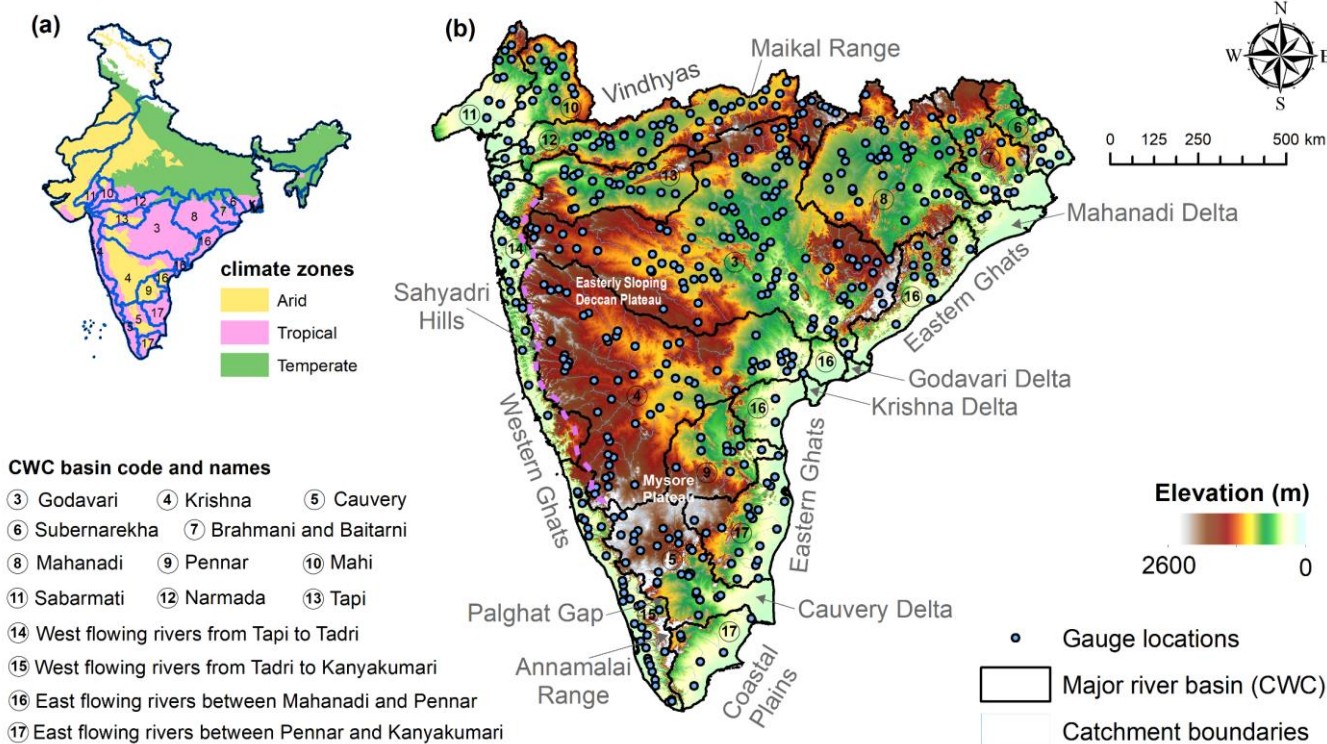

**Figure 1. (a)** Major river basins in Peninsular India, defined by the Central Water Commission (CWC), spread across various climate zones,
and **(b)** Geography of Peninsular India with major river basins, gauge locations, catchment boundaries, and elevation map.



Daily-scale streamflow and water level observations for Indian catchments are publicly accessible via the online portal India-WRIS (https://indiawris.gov.in/wris/#/DataDownload). Users can navigate the portal to select the data source (agencies providing river flow and water level data) and location (such as river basin and gauge name) to download river flow and water level data in Excel (.xlsx) format (**Fig. A1** and **A2**). Currently, India-WRIS imposes a maximum limit of one year for each

download. To obtain long-term time series, users must combine data by downloading one year at a time. This process can be tedious, but it is necessary to acquire river flow data for Indian catchments. Following this process, we compiled the available streamflow observations from 1 January 1980 to 31 December 2020 from India-WRIS and provided them in the CAMELS-INDIA dataset. Our preliminary analysis shows that most catchments have reliable data availability (less than 20% missing values for all hydrological years) from 1980 to 2018 (**Fig. 2**). However, it's worth noting that the India-WRIS portal was

launched in July 2019. Since then, continuous efforts have been made to digitize the available data and update the information on the portal. We anticipate that, with time, observations from the rest of the gauges will be made available for users to download. Therefore, we extracted catchment mean meteorological forcings and static attributes for all 472 catchments.

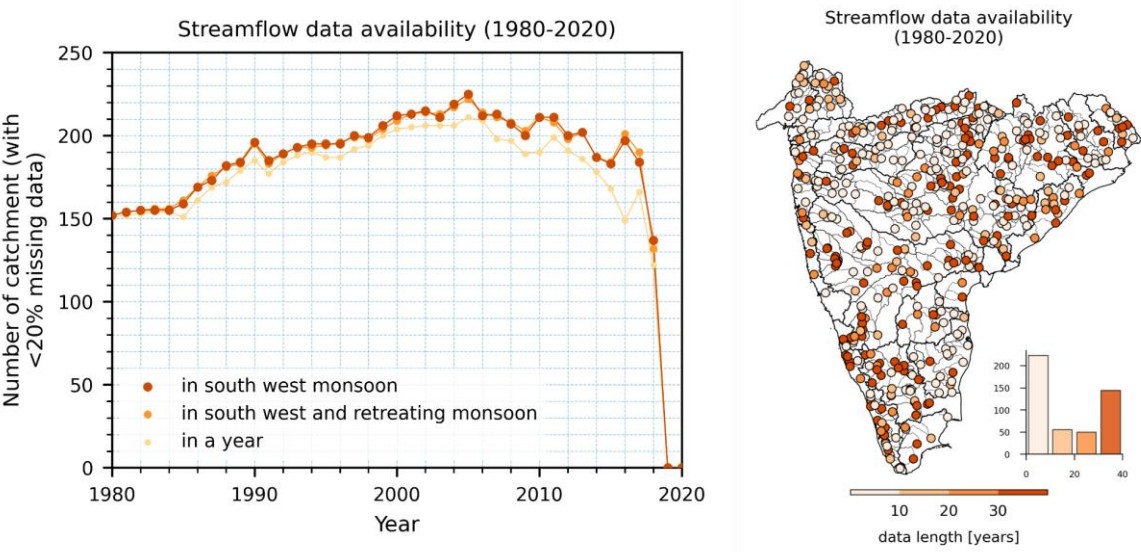

**Figure 2.** Streamflow data availability for each gauge station and the line plot with markers indicating the maximum number of catchments
have long-term flow data from 1980 to 2018. The indicated years are hydrological years (starting from 1 June).

## 4 Meteorological forcings

For large-sample studies, meteorological time series are often extracted from gridded datasets (Fowler et al., 2021). In CAMELS-INDIA, we extracted daily meteorological time series for 19 variables (listed in **Table A1**) from a nationwide gridded dataset covering the period from 1 January 1980 to 31 December 2020, spanning 41 years. We used gridded

precipitation (0.25° spatial resolution) (Pai et al., 2014) and maximum and minimum temperature (1° spatial resolution) (Srivastava et al., 2009) datasets from the IMD, which are the only available and widely utilized national dataset for India.





Daily time series for surface downward long-wave and short-wave radiation flux, U-component and V-component of wind (at 10 m), relative humidity (at 2 m), evaporation rates from the soil surface and canopy, and soil moisture at four different layers (0-0.1 m, 0.1-0.35 m, 0.35-1 m, and 1-3 m below ground) were extracted from the IMDAA dataset (Rani et al., 2021). This
dataset, with a resolution of approximately 12 km, is presently the highest-resolution gridded dataset available for the Indian monsoon region. As the actual and potential evapotranspiration (AET and PET) dataset over India is not available from national sources, we obtained a daily time series of AET and PET from the Global Land Evaporation Amsterdam Model (GLEAM) (Miralles et al., 2011). We also extracted daily time series of PET from Singer et al. (2021), which is presently the highest resolution (0.1°) gridded dataset developed using the ERA5-Land reanalysis dataset, to facilitate comparison. For all
meteorological variables, spatially averaged time series for each catchment were calculated using area-weighted averages for each day. The basin-wise meteorological time series is in a compressed zip file named "catchment_mean_forcings.zip" in the CAMELS-INDIA dataset (Mangukiya et al., 2024).

## 5 Catchment attributes

In CAMELS-INDIA, we compiled and calculated 211 catchment attributes representing location and topography, climate,
hydrological signatures, land-use land cover (LULC), soil and geology, and anthropogenic influences. **Table 1** summarizes the file names and descriptions of the attributes provided within the files in the CAMELS-INDIA dataset. In India, CWC has divided the entire country into 22 basins and provided a unique basin code for identification. In CAMELS-INDIA, we created a 5-digit gauge station identifier (the first two digits are CWC basin code, and the last three digits are station number) and used it as the gauge ID throughout the dataset. For each gauge ID, we provided the station's name as in the CWC database (CWC,
2021) and the name of the river/tributary and basin on which the station is located (**Table A2**). For ease of use, we also provided the GHI station ID and the GHI assigned group from the GHI dataset to associate the catchment attributes with the metadata provided in the GHI dataset (Goteti, 2023).

**Table 1.** Summary of 211 catchment attributes provided in CAMELS-INDIA

| File name | Attributes description |
|---|---|
| camels_India_name | 7 attributes (Table A2) representing gauge name and identifier |
| camels_India_topo | 16 attributes (Table A3) representing location and topography |
| camels_India_clim | 42 attributes (Table A4) representing climate indices |
| camels_India_hydro | 73 attributes (Table A5) representing hydrological signatures |
| camels_India_land | 13 attributes (Table A6) representing land cover characteristics |
| camels_India_soil | 28 attributes (Table A7) representing soil characteristics |
| camels_India_geol | 7 attributes (Table A8) representing geological characteristics |
| camels_India_anth | 25 attributes (Table A9) representing anthropogenic influence in the catchment |



## 5.1 Location and topography

The attributes representing the location and catchment area for each gauge ID are compiled from both the CWC and GHI datasets in CAMELS-INDIA (**Table A3**). However, it's worth noting that for many gauge stations, the CWC documented spurious gauge locations and catchment areas (Goteti, 2023). Therefore, we preferred the corrected locations provided within the GHI dataset for plotting the gauge locations in this manuscript. For topographic characteristics, elevation and slope are extracted using the 3 arcsec (~90 m) resolution Digital Elevation Model (DEM) of the Shuttle Radar Topography Mission

(SRTM) (Farr et al., 2007), as these are the key controlling factors of catchment behavior (Addor et al., 2017). The catchment areas range from 125.7 to 308433.8 km² , with quartile values of 1095.38 km² (first quartile), 3042.2 km² (second quartile), and 11990.63 km² (third quartile). **Figure 3a** shows the spatial distribution of the catchment area and highlights that there are 131 catchments with an area greater than 10,000 km$^2$. The average elevation becomes less meaningful for such large catchments due to spatial heterogeneity. Moreover, the west-flowing rivers from Tadri to Kanyakumari originate from the steep mountains

and meet the Arabian Sea, flowing through plain regions, resulting in lower average elevations and higher slopes (**Fig. 3b-c**). Therefore, we also computed minimum, maximum, and median catchment elevation and slope for all the gauges to represent the spatial heterogeneity of topographical features in the CAMELS-INDIA dataset. The catchment mean elevation ranges from 58.04 to 1687.24 m, with quartile values of 361.04 m (first quartile), 470.37 m (second quartile), and 617.9 m (third quartile), while the mean slope ranges from 1.07 to 32.15%, with quartile values of 4.11% (first quartile), 6.23% (second quartile), and

10.02% (third quartile). Additionally, the catchment mean drainage path slope is also estimated using SRTM DEM (**Fig. 3d**). The mean drainage path slope of the catchments ranges from 1.22 to 74.88 m/km, with mean and median slopes of 8.93 and 6.35 m/km, respectively. Overall, the topographic attributes show that the high-altitude catchments with moderate to steep slopes are located in the Western and Eastern Ghats regions, while the catchments in central India have gentler slopes.

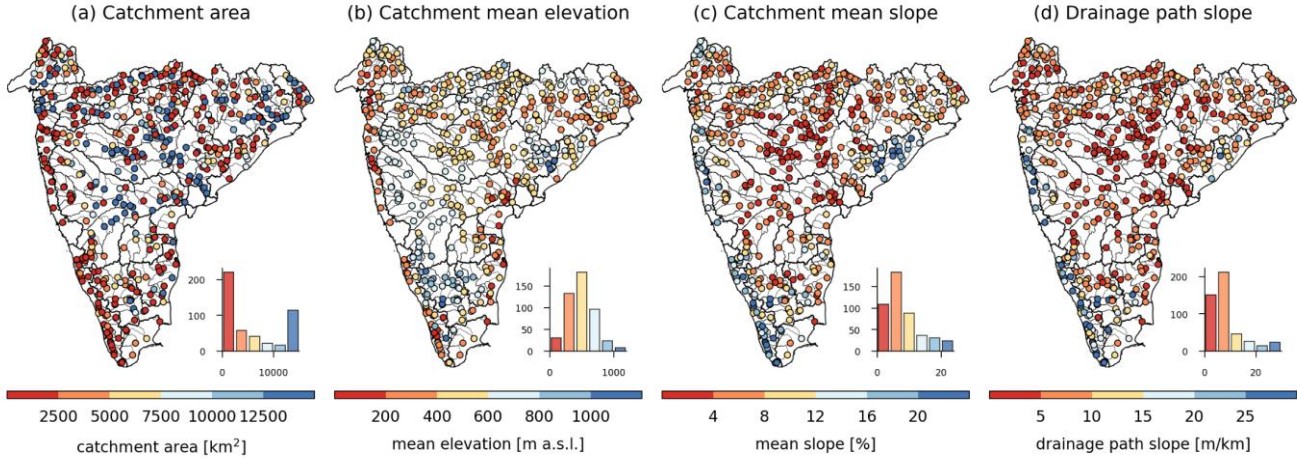

**Figure 3.** Topographic characteristics of catchments in Peninsular India. The histograms depict the frequency distribution of catchments across the bins. **(a)** catchment area in km$^2$, **(b)** catchment mean elevation in meters above mean sea level, **(c)** catchment mean slope in percentage, and **(d)** catchment mean drainage path slope in m/km.



## 5.2 Climate indices

We computed climate indices similar to Addor et al. (2017), which represent both mean and extreme events, using the
meteorological time series described in **Section 4**. Additionally, we calculated the monthly and annual precipitation variability,
precipitation uniformity, asynchronicity, and the maximum number of consecutive days of extreme event occurrence and their
timings (**Table A4**). To compute aridity, we used the ratio of mean annual precipitation over the PET, following the approach
of Addor et al. (2017). Moreover, we also derived the aridity index as the ratio of the deficit between potential and actual
evapotranspiration over PET. As an additional reference, we extracted the spatially averaged aridity index from Trabucco and
Zomer (2018). The precipitation uniformity indicates how uniformly the annual maximum precipitation is distributed across
the days of a year, and it is estimated by Relative Entropy, a metric proposed by Feng et al. (2013). A zero-precipitation
uniformity value indicates all the days have equal precipitation, whereas a value of 1 indicates that all the annual maximum
precipitation occurred in a single day (Dey and Mujumdar, 2019). The asynchronicity index measures the relative magnitude
and phase differences between long-term monthly precipitation and potential evapotranspiration (Feng et al., 2019). The
frequency of high precipitation days is estimated when the observed precipitation is at least five times the mean daily
precipitation. The frequency of low precipitation days is calculated when the observed precipitation is less than 1 mm/day.
The average consecutive days of high precipitation are used to estimate the average duration of high precipitation, and the
average consecutive dry days are used to estimate the average duration of low precipitation. The timing of high and low
precipitation is defined as the season (Monsoon – June, July, August and September; Pre-monsoon – January, February, March,
April and May; Post-monsoon – October, November and December) when most of the high and low precipitation events
occurs.

The spatial distribution of the selected climate indices is shown in **Figure 4**. The variability of annual precipitation and
frequency of high precipitation days are notably higher (with a coefficient of variation > 0.8 and more than 21 days with
precipitation ≥ 5 times mean daily precipitation) in the Mahi, Narmada, Pennar, Sabarmati, and Tapi basins (**Fig. 4a-b**). We
observed high precipitation events, mainly concentrated during the monsoon and post-monsoon seasons in the majority and
southern parts of the region, respectively. It highlights the dominance of the southwest monsoon (June to September) in the
region and the impact of the northeast monsoon in the southern part during winter (Das et al., 2022; Das and Jain, 2023). The
catchments along the western coast experience prolonged high precipitation in the southwest monsoon season and exhibit
evaporation rates of more than 1.2 mm/day (**Fig. 4c-d**). India has a seasonal precipitation pattern, with most precipitation
occurring during the southwest monsoon. Consequently, most Indian catchments experience more than 210 dry days in a year
(**Fig. 4e**). Moreover, the Mahi, Narmada, Sabarmati, and Tapi basins, along with the catchments of west-flowing rivers between
the Tapi and Tadri basins show extreme seasonality (Rai and Dimri, 2020), receiving most of the precipitation in 1-2 month,
resulting in the prolonged dry periods (**Fig. 4f-g**). The catchments along the southwest coast and eastern sides of Peninsular
India are relatively more humid compared to the catchments of Godavari, Krishna, Mahi, Pennar, Sabarmati, and Tapi basins
(**Fig. 4h**). A sharp transition in the aridity index is observed across the Western Ghats – highlighting the increased precipitation

on the leeward side and a decrease in the rain shadow region of the Western Ghats. A distinct north-south pattern in the asynchronicity index between long-term precipitation and PET is observed – with a strong out-of-phase relationship in the north and central parts of Peninsular India (**Fig. A3**). In contrast, an in-phase relationship is observed in the southern part of the region. In CAMELS-INDIA, we also provide mean indices for temperature, relative humidity, radiation flux, win speed,

and soil moisture to understand the climatic conditions over Peninsular India comprehensively. Higher mean daily precipitation is observed in the southern part of the region, and the precipitation decreases towards the central part of the region (**Fig. A3**). The northern and eastern parts of the region exhibit moderate precipitation. The spatial patterns of PET and AET are similar – moderate magnitudes are in the central and northern parts, and high values are in the southern part (**Fig. A3**).

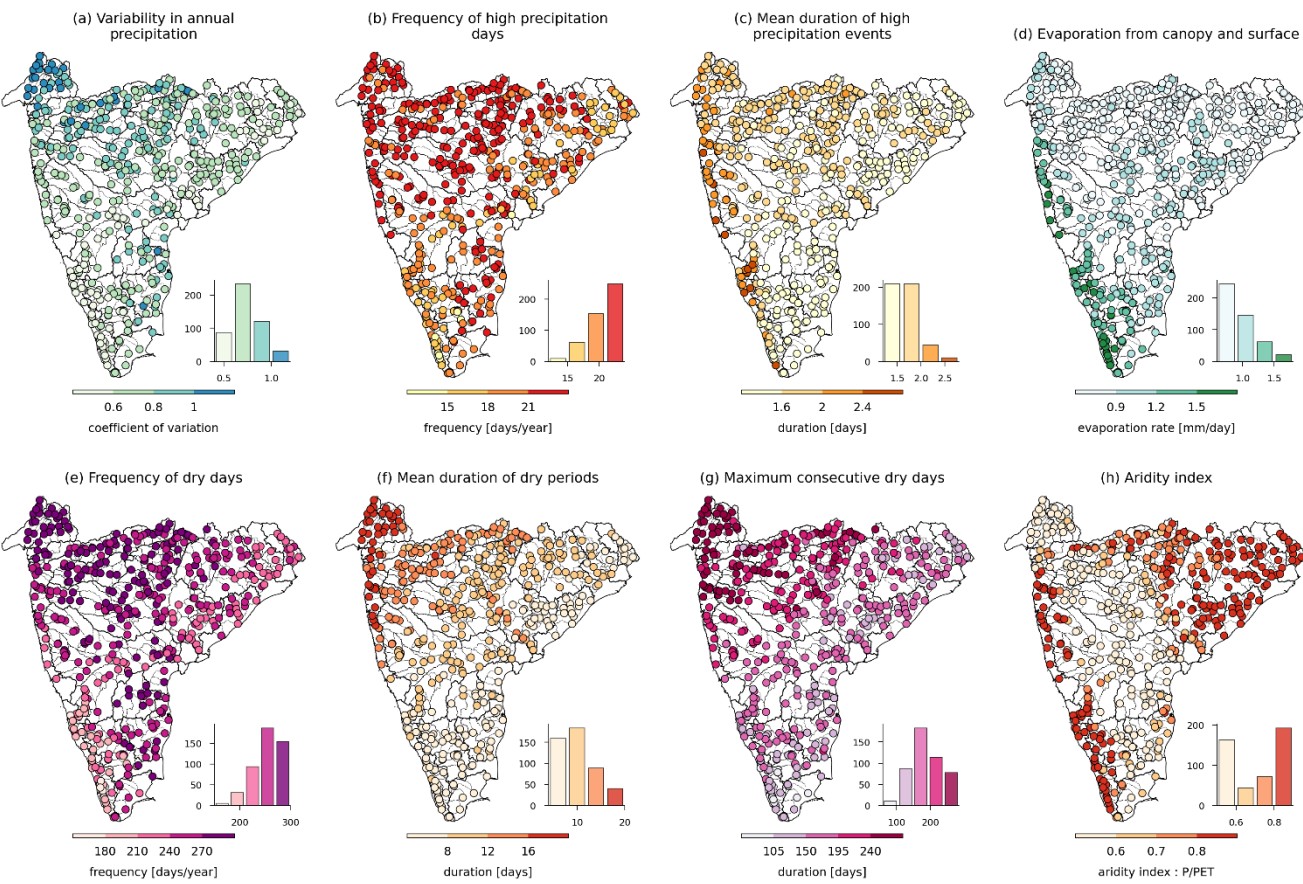

**Figure 4.** Climate indices for catchments in Peninsular India. The histograms depict the frequency distribution of catchments across the bins. **(a)** variation in annual precipitation patterns (higher values indicate more significant variation), **(b)** frequency of days with precipitation ≥ 5 times mean daily precipitation, **(c)** average number of consecutive days with precipitation ≥ 5 times mean daily precipitation, **(d)** mean daily evaporation rate from canopy and soil surface, **(e)** frequency of days with precipitation < 1 mm/day, **(f)** average number of consecutive days with precipitation < 1 mm/day, **(g)** maximum number of consecutive days with precipitation < 1 mm/day, and **(h)** aridity index (P/PET).



### 5.3 Hydrological signatures

To calculate hydrological signatures, we compiled available streamflow observations from IndiaWRIS, and indices were computed for gauges with at least 25 years of data and less than 20% missing values for all hydrological years (1 June to 31 May) between 1980 and 2020. The hydrological signatures representing the mean flow and extreme flow events are included in the CAMELS-INDIA (**Table A5**), similar to Addor et al. (2017). Additionally, due to seasonal precipitation patterns in India, we also computed seasonal flow and its variability, providing quartiles of flow for the southwest monsoon season. For this purpose, we also included gauges with available streamflow observations during specific seasons with less than 20% missing values for all months. In general, streamflow comprises two components – baseflow and quick flow. The baseflow index – the ratio of long-term baseflow to long-term total flow – is estimated using the method described in Ladson et al. (2013) available in the TOSSH toolbox (Gnann et al., 2021). The higher the baseflow index, the more the contribution of the baseflow to the total streamflow. The slope of the flow duration curve (FDC) is used to estimate the variability of streamflow. The slope of FDC is calculated as the slope of the curve between the log-transformed $33^{rd}$ and $66^{th}$ percentiles of daily streamflow over the period of observation (Chouaib et al., 2018; Yokoo and Sivapalan, 2011). A high slope value indicates highly variable streamflow due to pronounced streamflow seasonality or rapid response to precipitation events. Streamflow elasticity quantifies the sensitivity of mean annual precipitation (Sankarasubramanian et al., 2001). A value of streamflow elasticity $m$ indicates that there will be $m$% change in mean annual streamflow with respect to 1% change in mean annual precipitation. In addition, runoff ratio – the ratio of long-term mean daily flow to long-term mean daily precipitation – is estimated, which measures the fraction of precipitation that, on average, gets converted to streamflow. The streamflow uniformity – measured using Gini's coefficient (Gudmundsson et al., 2018), ranges from 0 to 1, where 0 indicates a uniform distribution of flows throughout the year, and 1 indicates that all the flows occur on a single day, with values between 0 and 1 representing intermediate cases. Apart from the measures of streamflow variability, attributes quantifying the behavior of extreme streamflow conditions are also quantified. The high-flow and low-flow thresholds during the observation period are computed based on the $95^{th}$ and $25^{th}$ percentile of the daily flows. Moreover, we computed approximately 40 indices of hydrological alterations, representing monthly water availability and variability, annual extreme events and their timing, and the frequency and rate of change in flow conditions. The primary limitations with the hydrologic signatures derived are: 1) many attributes can be associated with the size of the catchments, and 2) the causal factors of the extreme flow conditions are not considered.

The mean streamflow pattern closely follows the spatial patterns of the precipitation. The catchments of west-flowing rivers and Brahmani and Baitarni, Godavari, Mahanadi, Narmada, and Subernarekha basins exhibit higher flows (> 1 mm/day) throughout the year, including the southwest monsoon season (**Fig. 5a-b**). A high variation in the streamflow elasticity is observed in the arid regions, and less variation is observed in the humid regions (**Fig. A3**). However, the sensitivity of streamflow change to precipitation change is more in the arid regions. The streamflow uniformity is higher in the central, Eastern Ghats, and delta regions and smaller in the Western Ghats region (**Fig. A3**). High variability (with increased values)



in the baseflow index is observed in the southern region, whereas this variability tends to reduce in the central and the northern parts of the region (**Fig. A3**). In addition, the sensitivity of streamflow to precipitation decreases with increasing baseflow index – highlighting the role of baseflow in sustaining the flows. The catchments along the southwest coast have a high runoff ratio ($> 0.5$) and relatively low variability in daily flows (**Fig. 5c-d**). The majority of Indian catchments exhibit low-flows for 90 to 120 days during the summer season (March-May) with consecutive 30 to 60 days of low-flows (**Fig. 5e-f**). A similar pattern can be observed for high-flows but for a shorter duration, indicating the influence of the dams (**Fig. 5g-h**). Because most dams in India are operated to store water from high precipitation during the southwest monsoon season and gradually release it during summer for irrigation and other water demands. Evidence for this can be seen in **Figure 5i-k**, indicating a higher number of hydrological reversals ($> 100$ in a year) despite seasonal precipitation patterns.





**Figure 5.** Streamflow characteristics for catchments in Peninsular India. The histograms depict the frequency distribution of catchments across the bins. **(a)** mean daily streamflow, **(b)** mean daily streamflow of southwest monsoon season, **(c)** runoff ratio (q_mean/p_mean), **(d)** variability of daily streamflow, **(e)** mean consecutive low flow days (flow < 25th percentile daily flow), **(f)** frequency of low flows in a year, **(g)** mean consecutive high flow days (flow > 95th percentile daily flow), **(h)** frequency of high flows, **(i)** mean of the positive difference between consecutive flow values, **(j)** mean of the negative difference between consecutive flow values, **(k)** mean number of hydrologic reversals (i.e., number of peaks in hydrograph), and **(l)** mean annual flow volume.





## 5.4 Land cover characteristics

Land cover attributes were extracted from the Sentinel-2 10m LULC time series (Karra et al., 2021), providing eight LULC classes, including water, trees, flooded vegetation, crops, built-up areas, bare ground, snow cover, and rangeland. Due to the absence of snow cover in Peninsular Indian catchments, we excluded it, and the temporal average of the seven remaining LULC classes was extracted as fractions of the catchment (**Table A6**). Additionally, spatiotemporally averaged (from 2001 to 2020) minimum and maximum leaf area index (LAI) of the catchment were extracted from MCD15A2H MODIS/Terra+Aqua

Leaf Area Index/FPAR 8-day L4 Global 500m SIN Grid V006 [Data set] to represent the vertical density of vegetation. The maximum LAI will help set the boundary conditions of evaporation rates from the canopy and vegetation interception, while the difference between maximum and minimum LAI will represent the seasonal variation of LAI (Addor et al., 2017). The spatial variation of different LULC classes indicates that catchments of west-flowing rivers from Tadri to Kanyakumari, east-flowing rivers between Pennar and Kanyakumari, and the Cauvery basin have higher (> 8%) urban areas. In comparison, more than

than 50% of the catchment areas of the Godavari, Krishna, Mahi, Narmada, and Tapi basins are covered with agricultural land (**Fig. 6a-b**). To meet agricultural water demands in these catchments, several large and medium reservoirs and lakes are present in this region, covering more than 2% of the catchment area (**Fig. 6c**). The catchments along the southwest coast are mainly covered with trees and exhibit lower seasonal variability of LAI (**Fig. 6d-f**). On the other hand, catchments in central India show high seasonal variability of LAI, primarily due to seasonal crops, as these catchments have a very low fraction of forest

cover (**Fig. 6d-f**).

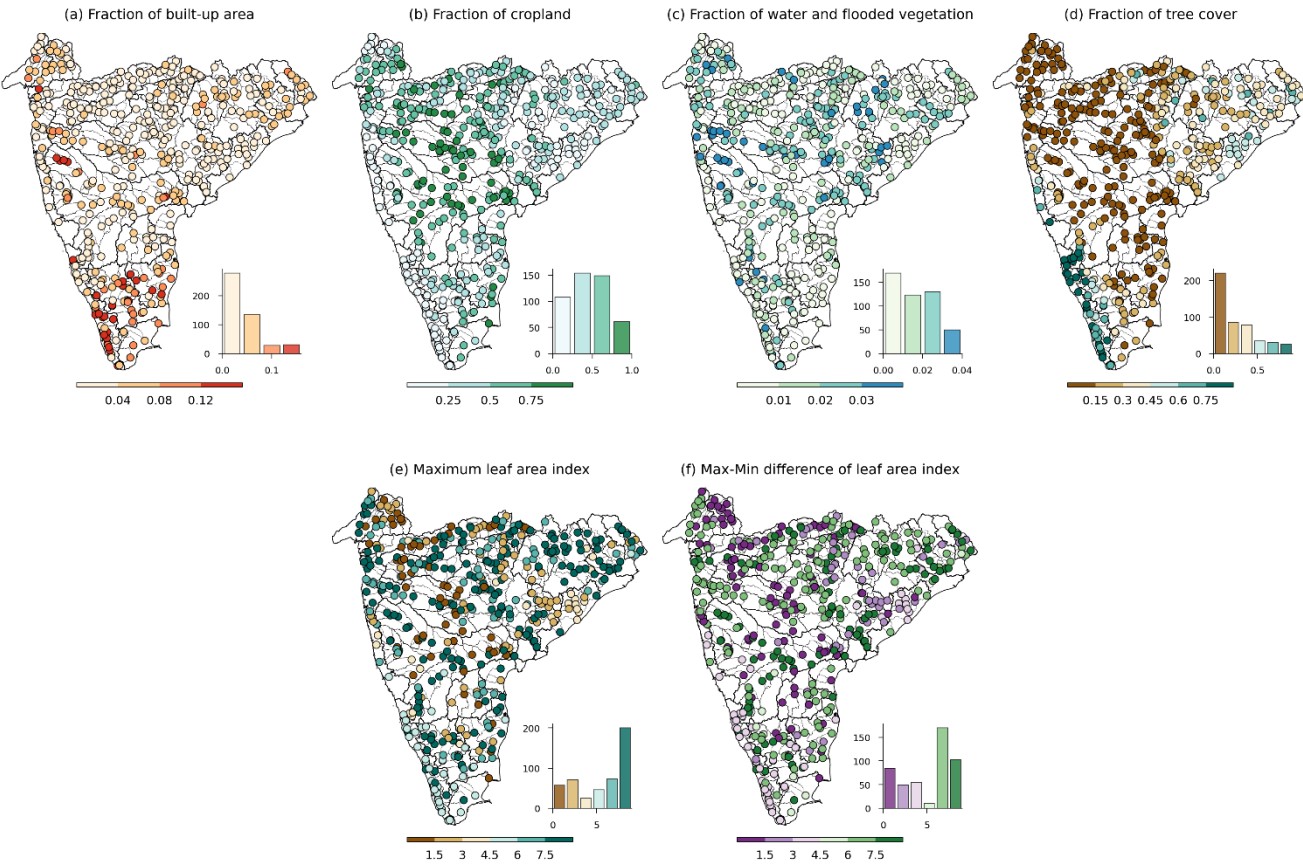

**Figure 6.** Land-use land cover characteristics for catchments in peninsular India. The histograms depict the frequency distribution of catchments across the bins. **(a-d)** fraction of built-up area, cropland, water and flooded vegetation, and tree cover, respectively, **(e)** maximum leaf area index, and **(f)** difference between maximum and minimum leaf area index.

### 5.5 Soil and Geological characteristics

The attributes related to soil characteristics of the catchment were derived from global data sources (**Table A7**), as national datasets related to soil characteristics are either not openly available or not in digitized form at present. The average soil depth of the catchments was extracted from Pelletier et al. (2016), which provides the thickness of soil and sediment deposits with a 30 arcsec resolution. The saturated hydraulic conductivity, available water storage, and fraction of organic matter content for both topsoil (0-30 cm) and subsoil (30-200 cm) were extracted from HiHydroSoil v2 at 250m resolution (Simons et al., 2020). The available water storage capacity of the soil was extracted from Food and Agriculture Organization (FAO) soil data (Fischer et al., 2008). The fraction of sand, silt, clay, and gravel, bulk density of soil, and organic carbon content in soil for both topsoil (0-30 cm) and subsoil (30-100 cm) were extracted from the Harmonized World Soil Database v2.0 (FAO and IISA, 2023). The catchment mean annual average water table depth was extracted from Fan et al. (2013). Additionally, we also extracted



the major hydrologic soil group (HSG) from the HiHydroSoil v2 (Simons et al., 2020). The HSG helps derive the runoff curve
number utilized in hydrological modeling for direct runoff estimation.

The spatial variability of soil attributes shows that the catchments of the Mahanadi and lower Godavari basins have a high
fraction of sand, while catchments along the southwest coast, Brahmani and Baitarni, Sabarmati, and Subernarekha basins

have a high fraction of silt and gravel (**Fig. 7a-d**). The catchments of the Krishna, Narmada, Tapi, and upper Godavari basins
have a high fraction of clay in the soil. Catchments of west-flowing rivers from Tapi to Tadri have more than 4% organic
carbon content (**Fig. 7e**). Out of 472 catchments, 320 catchments in India have a soil depth up to 2 m, while 87 catchments,
mainly located along the east coast, lower Godavari, lower Narmada, and Sabarmati basins, exhibits soil depth of more than 3
m (**Fig. 7f**). The catchments located in the upper part of Peninsular India, mainly in the Godavari, Mahanadi, Mahi, Narmada,

Sabarmati, and Tapi basins, have a low available water storage capacity of the soil, in comparison to the catchments along the
southwest coast (**Fig. 7g**). The spatial variability of soil conductivity shows that catchments of lower Krishna, Mahi, Pennar,
and Sabarmati basins have high soil conductivity (> 5 cm/day) (**Fig. 7h**), and catchments of Peninsular India have moderate
to high runoff potential.

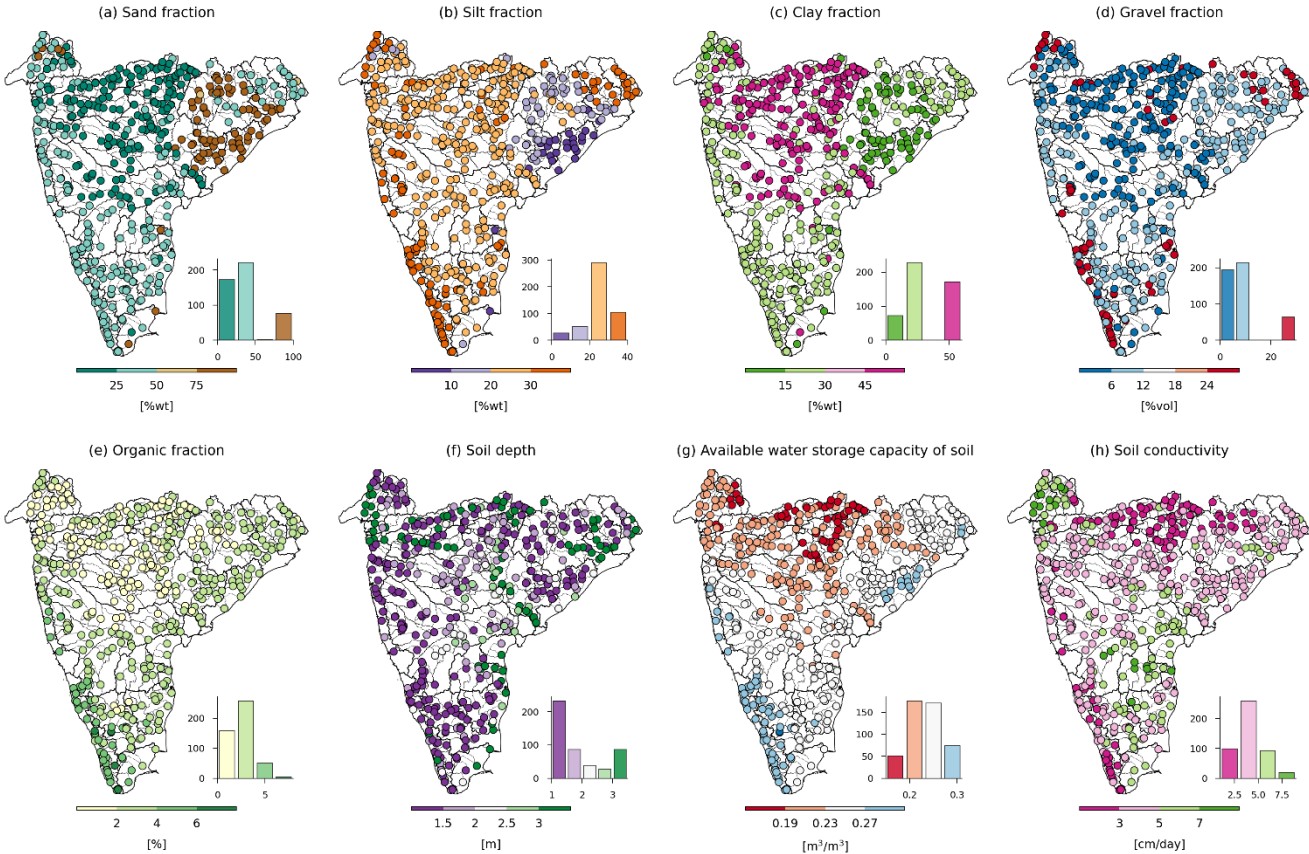

**Figure 7.** Soil characteristics for catchments in peninsular India. The histograms depict the frequency distribution of catchments across the
bins. **(a-e)** fraction of sand, silt, clay, gravel, and organic matter content in topsoil (0-30 cm), respectively, **(f)** average thickness of soil and





sedimentary deposit, **(g)** available water storage capacity of topsoil (0-30 cm), and **(h)** mean saturated hydraulic conductivity of topsoil (0-30 cm).

The geological attributes (**Table A8**) were computed following Addor et al. (2017). The first and second most common geological classes, their respective proportions within the catchment, and the fraction of 'carbonate sedimentary rocks' were extracted from the Global Lithological Map (GLiM) (Hartmann and Moosdorf, 2012). The mean subsurface porosity and permeability of the catchment were derived from the GLobal HYdrogeology MaPS (GLHYMPS) (Gleeson et al., 2014). The spatial variability of subsurface porosity and permeability indicates that catchments in the Narmada and Sabarmati basins have relatively high porosity (> 0.1), while those in the Mahi, Narmada, Tapi, upper Godavari, and upper Krishna basins exhibit high permeability (> 0.73 m$^2$) (**Fig. 8a-b**). The southern parts of the Peninsular region consist of the hard rock aquifer system with low porosity and permeability. The Peninsular region is the oldest and largest geomorphic province of India. There are seven dominant geological classes identified in the Peninsular region – basic volcanic, metamorphic, acid plutonic, siliciclastic sedimentary rocks, mixed sedimentary rocks, carbonate sedimentary rocks, and subordinate unconsolidated sediments. Out of 472 catchments, 179 have 'basic volcanic rocks' and 176 have 'metamorphic rocks' as the most common geological classes, with the majority of them having only a single geological class for the entire catchment (**Fig. 8c-g**). The rock types that are classified under basic volcanic rocks are basalts, tephrites, tholeiites, and lamprophyres (Hartmann and Moosdorf, 2012). The metamorphic class constitutes a variety of rocks from shales to gneiss, from amphibolite to quartzite. The groundwater movement in these two dominant classes is controlled by rock fractures and their continuities, depth of weathering, topography, nature, and size of recharge and discharge areas.



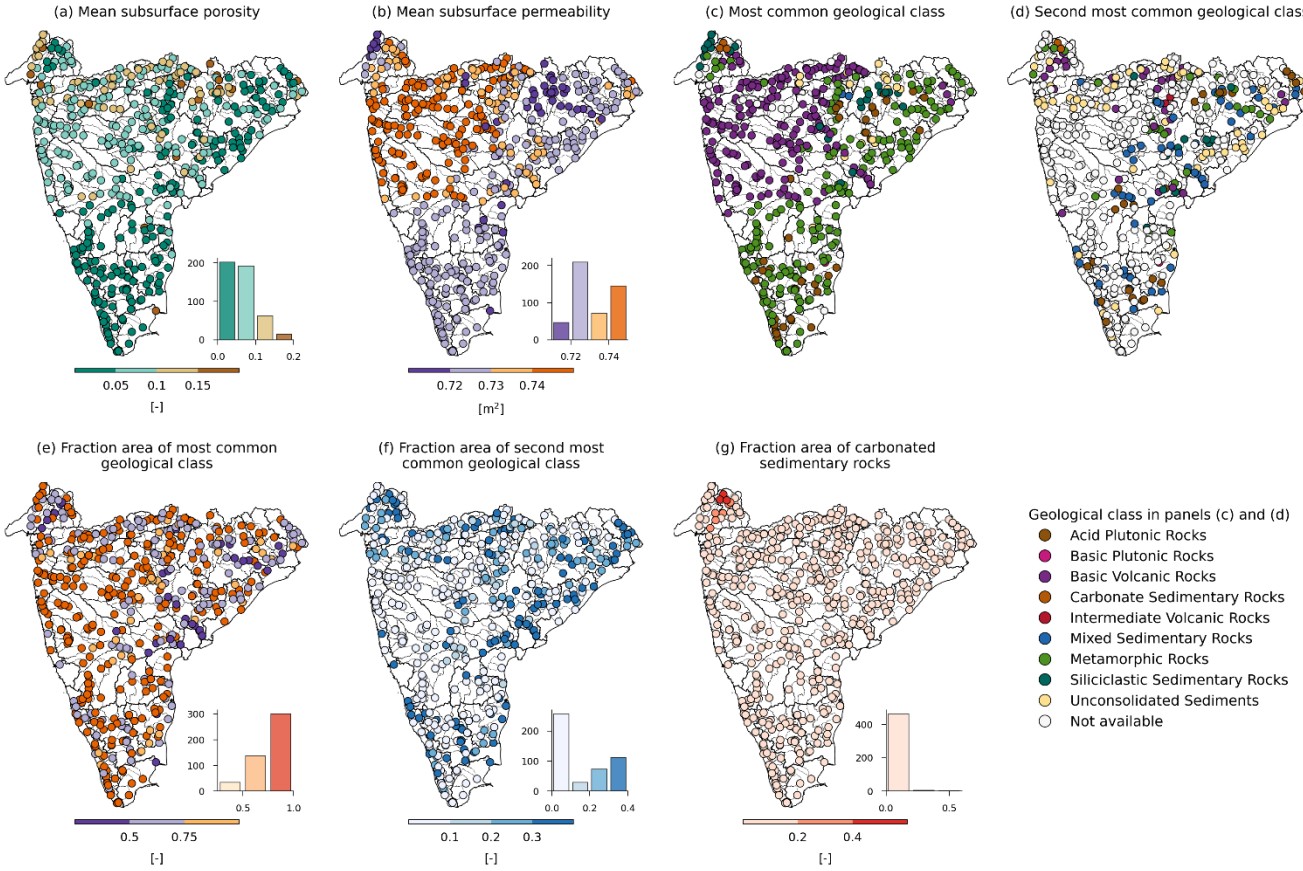

**Figure 8.** Geological characteristics for catchments in peninsular India. The histograms depict the frequency distribution of catchments across the bins. **(a-b)** mean subsurface porosity and permeability, **(c-d)** most common geological classes, **(e-f)** fraction of catchment area associated with most common geological classes, and **(g)** fraction of catchment area characterized as carbonated sedimentary rocks.

### 5.6 Anthropogenic influences

Catchments in India have varied degrees of anthropogenic influence. Due to seasonal rainfall patterns, water demands in the region are primarily met by several dams. In CAMELS-INDIA, the degree of human intervention within the catchments is quantified through the information on the number of dams, year of construction of the first and the recent dam, total cumulative storage of dams, and fraction of these storages used in hydropower generation, flood control, irrigation, drinking, flood storage, and navigation (**Table A9**). In addition, the reservoir index – a ratio of total storage volume to multiyear annual streamflow – is also estimated. The attributes of the number of dams in each catchment and their cumulative storage volume were extracted and digitized from India-WRIS and the Global Reservoir and Dam Database (GRanD) (Lehner et al., 2011). The water stored in these dams is mainly used for urban and agricultural purposes. Since quantitative measurements of water demands are unavailable, we included decadal population density data (WorldPop and CIESIN, 2018) and the fraction of urban areas and cropland (Roy et al., 2015) as indirect measures in CAMELS-INDIA.

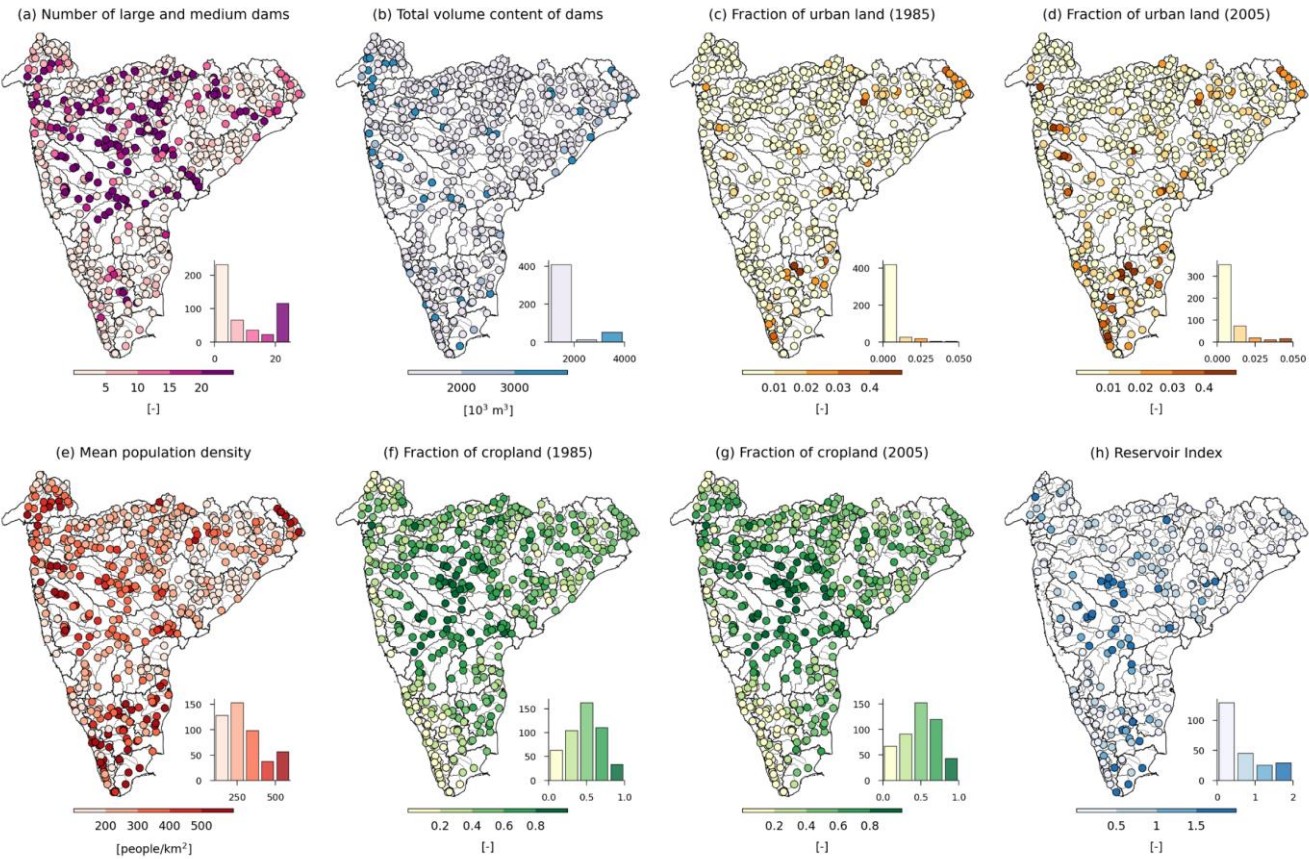

390

**Figure 9.** Attributes representing anthropogenic influences for catchments in Peninsular India. The histograms depict the frequency distribution of catchments across the bins. **(a)** total number of large and medium dams, **(b)** total volume content of dams, **(c-d)** decadal fraction of urban land cover in 1985 and 2005, respectively, **(e)** mean population density, **(f-g)** decadal fraction of cropland in 1985 and 2005, respectively, and **(h)** reservoir index.

395    The spatial distribution of large and medium dams across catchments shows significant regulation in the catchments of Cauvery, Godavari, Krishna, Mahanadi, Mahi, Narmada, and Tapi basins (**Fig.9a-b**). The number of dams within the catchment ranges from 0 to 1277, with a mean and median of 75 and 9, respectively. The total storage capacity ranges from 0 to 59929 Mm$^3$ with mean and median storage capacity of 3796 Mm$^3$ and 246 Mm$^3$, respectively. The decadal variation of the urban land cover and population density reveals a notable increase in urbanization within the catchments of southern India

400    from 1985 to 2015 (**Fig. 9c-e**). Conversely, the fraction of agricultural land remains relatively constant over the same period (**Fig. 9f-g**). The reservoir index, indicating the impact of dams on streamflow, is higher in Godavari, Krishna, and Cauvery basins whereas most of the catchments in Narmada basin and Western Ghats region have lower values of reservoir index (**Fig. 9h**). It is observed that majority of the dams in this region is served for irrigation purpose, whereas the dams in the southern part of the Peninsular region is mainly used for hydroelectric generation.





## 6. Regionally trained LSTM-based hydrological model for streamflow prediction

We used a Long-Short Term Memory (LSTM)-based regional hydrological model applied to Indian catchments by Mangukiya et al. (2023) to predict daily streamflow for all 472 catchments. The LSTM model architecture includes an input gate, output gate, forget gate, and a memory cell, which enables the model to learn long-term dependencies within the input datasets (Hochreiter and Schmidhuber, 1997). We trained the LSTM-based regional hydrological model using daily meteorological time series and catchment attributes as input and predicted daily streamflow (Mangukiya et al., 2023; Mangukiya and Sharma, 2024). The input data included daily meteorological time series of precipitation, maximum and minimum temperature, solar radiation, wind speed, and relative humidity, along with catchment attributes representing topographic, land cover, soil, and geological characteristics. The LSTM model was trained using a dataset from 159 catchments, ensuring a minimum data length of 28 years was available for each catchment between 1980 and 2020. The optimized hyperparameter values for the LSTM model were adopted from Mangukiya et al. (2023). The model was trained from 1 January 1991 to 31 December 2015, validated from 1 January 1980 to 31 December 1990, and tested from 1 January 2016 to 31 December 2020. In addition to 159 catchments, we tested the LSTM model's generalization capability to make streamflow predictions in 17 pseudo-ungauged catchments, which were held out during training.

The results indicate satisfactory model performance, with a median Nash-Sutcliffe Efficiency (NSE) of 0.59 and 0.57 during the test and validation periods, respectively (**Fig. 10a**). Notably, the LSTM model achieved a median correlation of 0.8, percentage bias of -7.64, Kling-Gupta Efficiency (KGE) of 0.62, root mean squared error (RMSE) of 121.5 m$^3$/s, low flow (bottom 30% of flow, FLV) bias of -1.4%, and high flow (top 2% of flow, FHV) bias of -15.72% (**Fig. 10b**). Additionally, we calculated the average RMSE of the observed and predicted flow duration curve (fdcRMSE) as an additional evaluation metric. The model achieved a median fdcRMSE of 127.09 m$^3$/s. However, we observed that the LSTM model performed poorly in challenging catchments, such as those with a high number of dams, non-perennial catchments, and catchments in arid and semi-arid climate zones. More details on model performance and limitations can be found in Mangukiya et al. (2023). The LSTM-based regional hydrological model, trained on 159 catchments, was used to predict daily streamflow for all 472 catchments from 1 January 1980 to 31 December 2020. Within CAMELS-INDIA, gauge-wise predicted streamflow series are provided in a compressed zip file named LSTM_pred_streamflow.zip. This predicted streamflow series is included in the dataset to support deep learning or machine learning-based hydrology research and can be used as a benchmark or baseline model for developing and testing hydrological models.



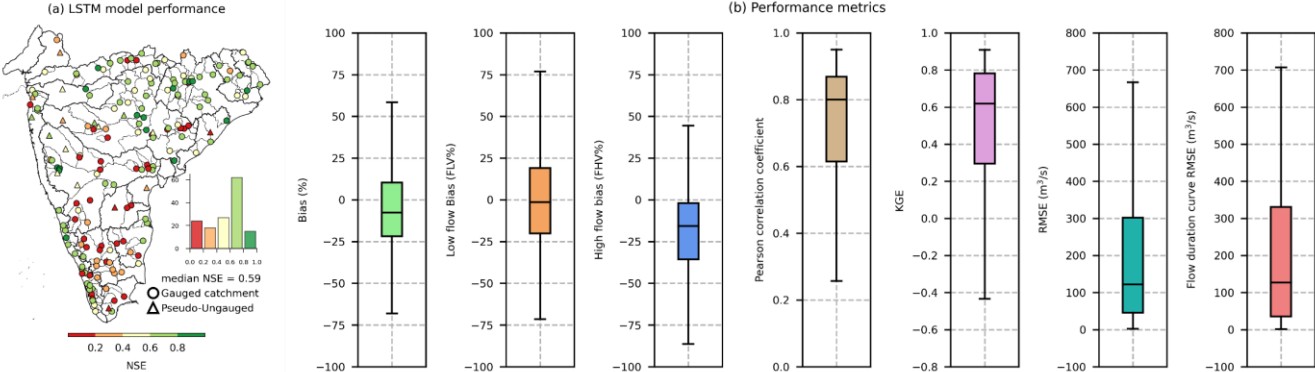

**Figure 10.** LSTM model performance. **(a)** spatial distribution of NSE, and **(b)** performance metrics.

## 7. Preliminary assessment of dataset quality and uncertainty

The preliminary assessment presented here focuses only on the catchment mean meteorological time series provided in CAMELS-INDIA. The precipitation and maximum and minimum temperature time series were extracted from the IMD dataset. The precipitation dataset from IMD is based on observation of rainfall from 6995 rain-gauge stations across India, and it accurately represents the spatial distribution of rainfall (Pai et al., 2014). The temperature dataset from IMD is based on 395 quality-controlled observatories across India. Observations from these stations are converted into a gridded product with a

spatial resolution of 1° x 1° using Shepard's angular distance weighting method (Srivastava et al., 2009). Goteti (2023) has provided a detailed comparison of the annual precipitation series of the catchment extracted from IMD and ECMWF Reanalysis (ERA). The results indicated a Pearson correlation coefficient greater than 0.75 for 31 catchments and between 0.5 to 0.75 for 331 catchments out of 472. The lower correlation was found only in the in the hilly terrain of the southwestern part of Peninsular India. Moreover, Mahto and Mishra (2019) also observed a general consistency between the ERA and IMD

datasets. We extracted the meteorological time series of solar radiation, wind speed, relative humidity, and soil moisture from the IMDAA data. IMDAA data is a high-resolution regional reanalysis of India, developed by Weather and Climate Modelling under the Ministry of Earth Sciences, India, with increased reliability and accuracy (Rani et al., 2021; Ashrit et al., 2020). **Figure 11a** shows the Pearson correlation of catchment mean annual time series for long-wave and short-wave solar radiation, as well as wind speed, extracted from IMDAA and those derived from the Global Land Data Assimilation System (GLDAS)

(Rodell et al., 2004). For the majority of catchments, the long-wave solar radiation extracted from the IMDAA dataset shows consistency with that of the GLDAS dataset, indicated by a high correlation coefficient. However, a lower correlation was observed for short-wave solar radiation, particularly in the hilly terrain of the Mysore Plateau and southern catchments. Similarly, wind speed also exhibited discrepancies in a few catchments in the southern region and in the catchments of the upper Eastern Ghats, lower Godavari, Mahanadi, and Brahmani and Baitarni basins. These discrepancies between the data

sources could be attributed to the different boundary conditions and forcings that are used to simulate the climate models



(Rodell et al., 2004; Rani et al., 2021). While global reanalysis products provide a convenient data source, their relatively coarse resolution (e.g., 25 km grid spacing) limits their ability to accurately capture climate variations in mountainous regions. In contrast, IMDAA, with its 10 km resolution, provides a more detailed representation of such variations. As demonstrated by Nayak et al. (2018), reanalysis products derived using Indian-specific boundary conditions and land-use data showed better

performance in capturing meteorological patterns in hilly areas compared with GLDAS.

To further evaluate the quality of the meteorological time series provided within the CAMELS-INDIA, we used it as input to the LSTM-based regional hydrological model (described in **Section 6**) and compared the model's performance with that of GLDAS meteorological time series as input for approximately 200 catchments with continuous streamflow observations from 1991 to 2015. The results indicate superior model performance when using IMDAA forcings as input compared to GLDAS

forcings (**Fig. 11b**). For the majority of the catchments (165 out of 200), the model performed better with IMDAA forcings. Minor improvements (with a NSE difference of ≈0.02) were observed in 13 catchments, while performance significantly deteriorated in 22 catchments with GLDAS forcings. Notably, the simulated streamflow based on IMDAA forcings outperformed that based on GLDAS forcings, with a median percentage bias of -11.74%, low flow bias (FLV) of -19.48%, and high flow bias (FHV) of -18.58%, compared to -20.39%, -22.5%, and -25.22%, respectively (**Fig. 11c**). Overall, the

preliminary assessment of the dataset suggests that the meteorological time series extracted from the IMD and IMDAA are the best available national data sources for Indian region, providing reliable model performance compared to global data sources. The preliminary results clearly demonstrated the CAMELS-INDIA dataset's potential to significantly enhance the performance of hydrological applications. However, it is crucial to acknowledge that the dataset is not without its limitations. Several factors, including data collection methods, processing techniques, and measurement errors, can introduce uncertainties into

the dataset. For instance, the use of diverse instruments and methodologies over time can lead to inconsistencies in measurements, particularly for variables like rainfall and streamflow. While the dataset provides catchment-average indices and series, the spatial resolution disparities between satellite, ground-based, and re-analysis products have a relatively limited impact on overall data quality. Nonetheless, gaps in data coverage and the presence of spurious values can further exacerbate the uncertainty. A detailed assessment and quantification of uncertainty is beyond the scope of this paper and will be addressed

in future versions of the dataset when ground-based observations become available for public use.



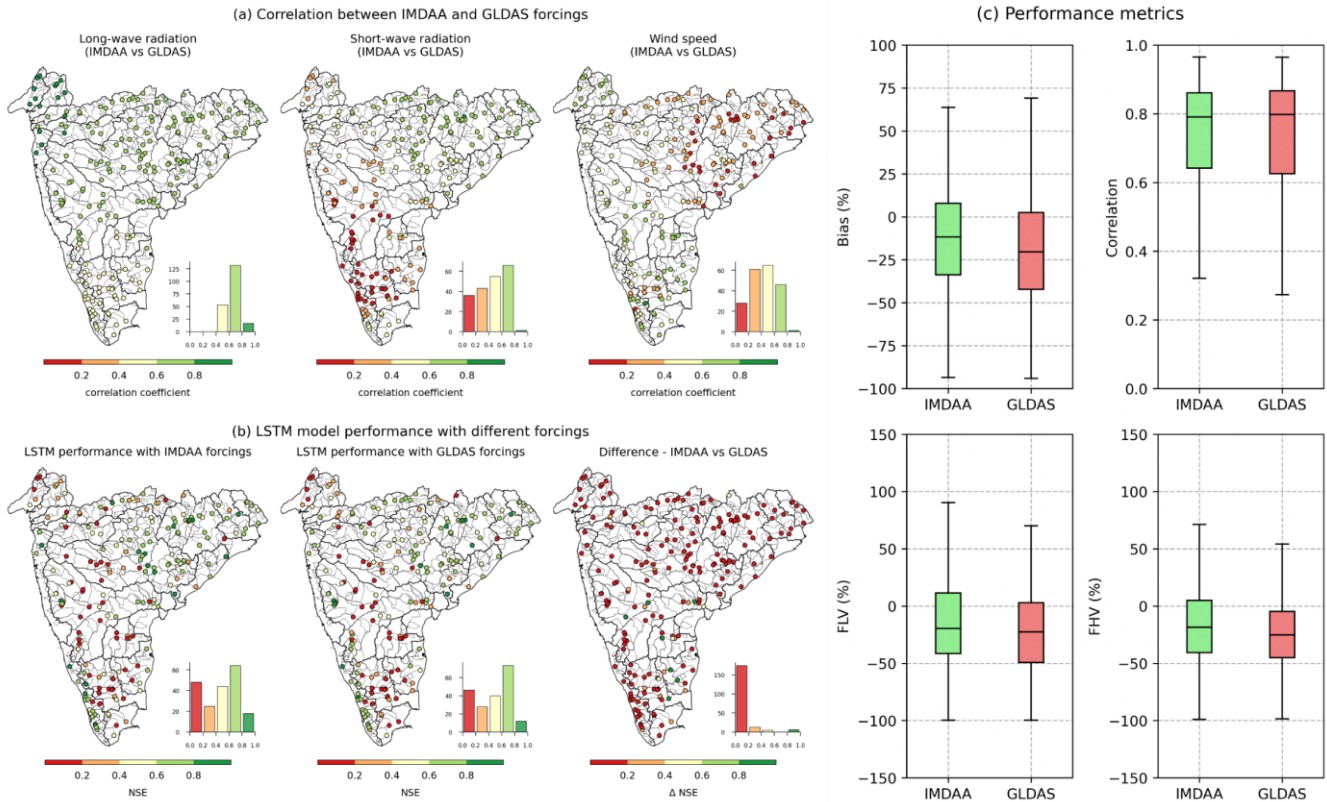

**Figure 11. (a)** Pearson correlation coefficient between catchment mean annual time series from IMDAA and GLDAS, **(b)** LSTM model performance with IMDAA and GLDAS as inputs, and **(c)** Calculated performance metrics of the LSTM model with IMDAA and GLDAS forcing inputs.

## 8. Possible future extensions

The CAMELS-INDIA dataset currently provides hydrometeorological time series and catchment attributes for only 472 catchments in peninsular India due to the availability of openly accessible and quality-controlled datasets. However, India has 4824 catchments at present, many of which are restricted due to their location in transboundary river basins. While the meteorological data are available for the entire country, the primary constraint is the availability of consistent streamflow observations. In future versions, we aim to address this limitation by applying the conceptual or physics-based and regionally trained LSTM-based hydrological model to other locations, thereby providing simulated streamflow series for catchments currently classified as restricted. Additionally, we intended to leverage the streamflow series from other sources, such as GloFAS (Harrigan et al., 2020) and satellite altimetry (Verma et al., 2021; Rai et al., 2021) to improve the spatial coverage of the dataset. To ensure accuracy, these streamflow time series will be validated against ground-based measurements at selected stations in restricted regions.

Groundwater is a vital component of understanding hydrological extremes such as floods (Sharma and Mujumdar, 2024) and droughts (Hellwig et al., 2020). However, the current CAMELS-INDIA dataset is limited by the absence of ground water data. To address this limitation, we aim to incorporate the ground water level data available at India-WRIS Portal (https://indiawris.gov.in/wris/#/groundWater) and derived ground water level data from Gravity Recovery and Climate

Experiment (GRACE) (Li et al., 2019; Moudgil and Rao, 2023; Gautam et al., 2024). The inclusion of ground water data will significantly improve the dataset's ability to capture the complex interactions between surface and sub-surface systems, thereby enhancing our understanding of hydrological processes and extreme events.

**Data availability**

The CAMELS-INDIA dataset is freely available at https://doi.org/10.5281/zenodo.13221214 (Mangukiya et al., 2024), which

includes: (1) '00_camels_India_data_description.pdf' file for description of data source and reference, and file structure of dataset in the repository, (2) 'attributes_csv.zip' and 'attributes_txt.zip' files containing all static catchment attributes in CSV and TEXT format, (3) 'catchment_mean_forcing.zip' file containing catchment mean meteorological time series for each catchment, (4) 'shapefiles_catchment.zip' file containing GIS shapefiles of catchments and gauge locations, and (5) 'streamflow_timeseries.zip' file containing available observed and LSTM-based hydrological model predicted streamflow

time series for all catchments.

**Concluding remarks**

India has hydrologically distinct catchments, each with unique characteristics. However, Indian catchments are often underutilized in global hydrological studies due to insufficient analysis-ready datasets. To address this gap, we introduce CAMELS-INDIA (Catchment Attributes and MEteorology for Large-sample Studies – India), which provides catchment mean

time series of meteorological variables and around 211 catchment attributes representing location and topography, climate, hydrological signatures, land-use land cover (LULC), soil and geology, and anthropogenic influences for 472 catchments in peninsular India. Such a dataset is essential for understanding hydrologic processes over multiple Spatiotemporal scales and various other applications for planning and regulating water resources in India. The CAMELS-INDIA follows the same standards of the previously developed CAMELS datasets for the USA, Chile, Brazil, Great Britain, Australia, Switzerland,

and Germany to facilitate comparisons with catchments of those countries and inclusion in global hydrological studies. CAMELS-INDIA serves as a stepping stone to provide large-sample hydrometeorological time series and attributes of the Indian catchments to the global and national hydrological community, and we plan to update and expand the dataset with additional catchment attributes and meteorological forcings as new national data sources become available. For example, future versions of CAMELS-INDIA could include additional catchment attributes to better characterize heterogeneity and regulations



within each catchment. Additionally, since data uncertainties are inherent, future studies will explore this through comparisons
       with additional data sources.

       The creation of CAMELS-INDIA aims to foster large-sample hydrological studies in India and promote the inclusion of Indian
       catchments in global hydrological research. Furthermore, it will enhance the reproducibility and transparency of hydrological
       studies in India by providing a standardized dataset.

**Appendices**

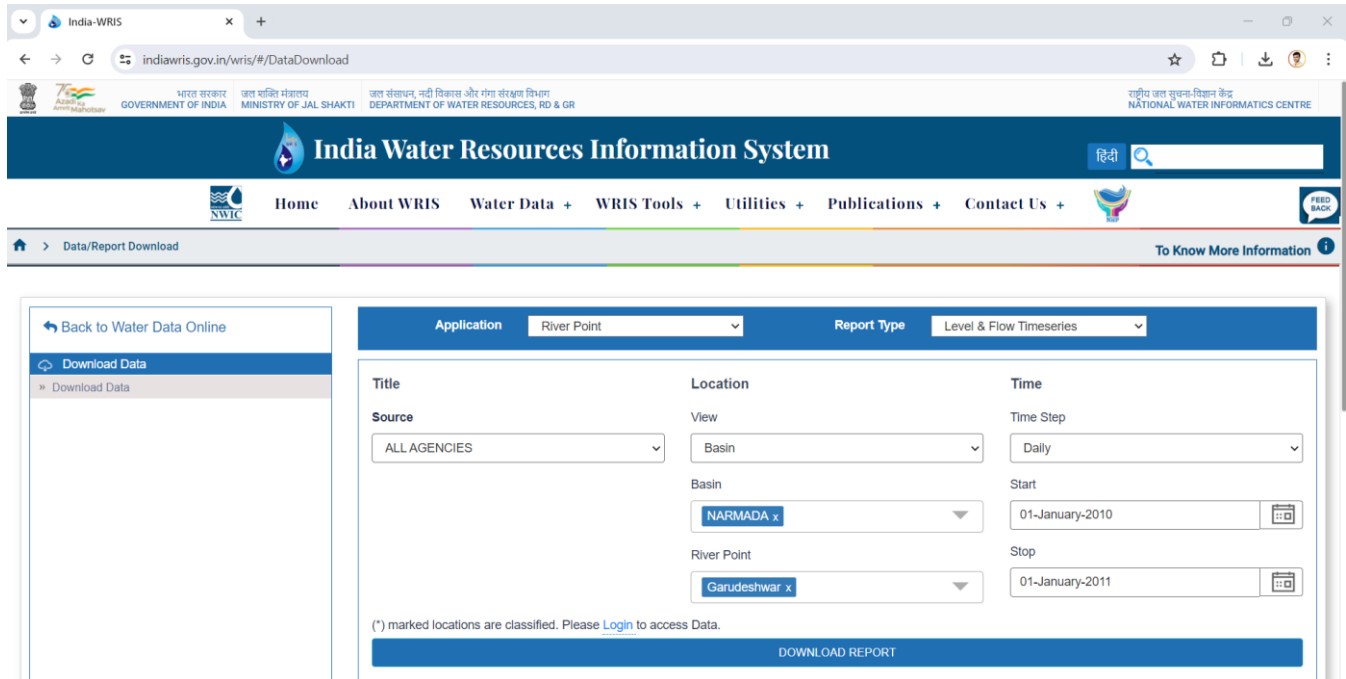

**Figure A1.** A snippet of the India-WRIS portal for obtaining streamflow observations. Users can select data sources, river basins, and station names to download data in Excel (.xlsx) format.

**Figure A2.** A snippet of the raw streamflow data downloaded for the 'Garudeshwar' gauge station for 2010.



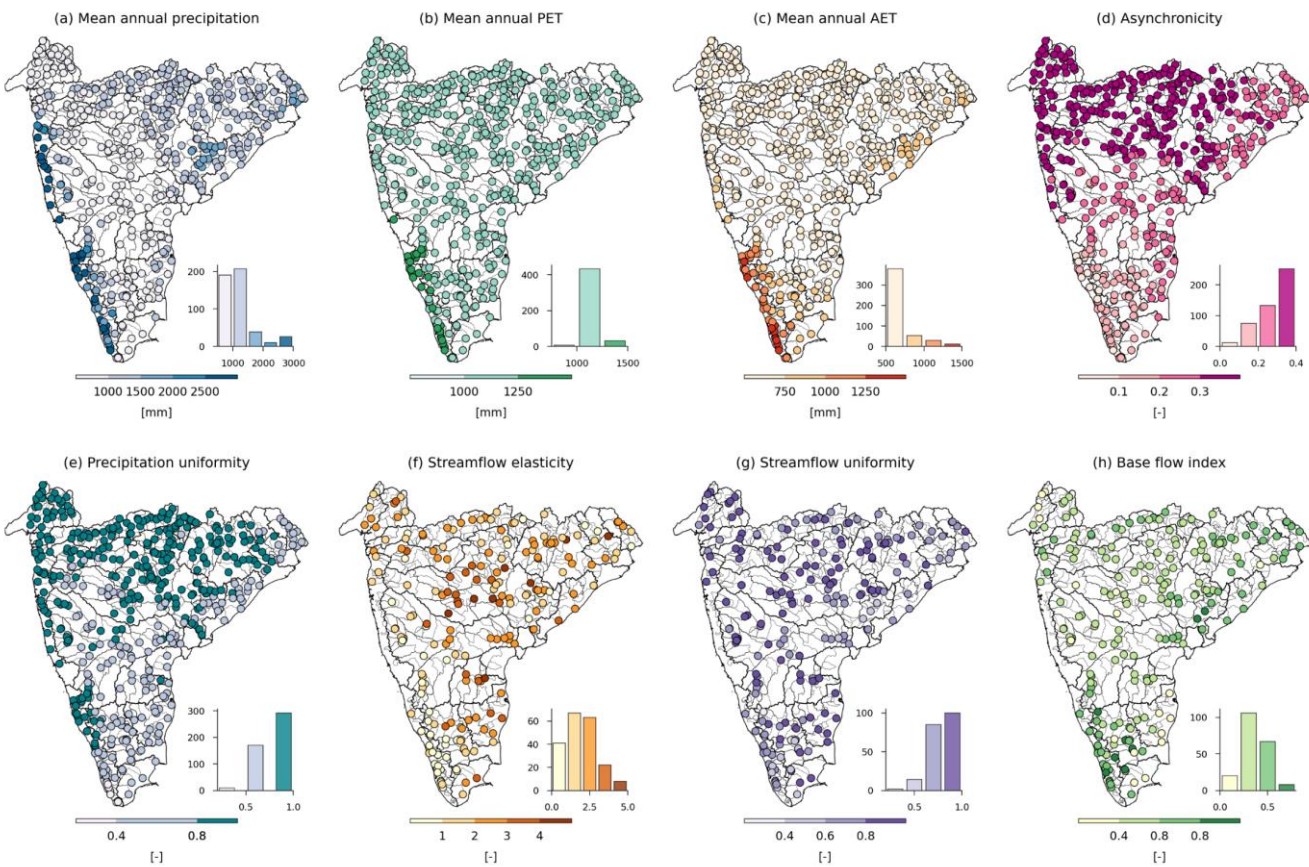

**Figure A3. (a)** mean annual precipitation, **(b)** mean annual actual evapotranspiration, **(c)** mean annual potential evapotranspiration, **(d)** asynchronicity, **(e)** precipitation uniformity, **(f)** streamflow elasticity, **(g)** streamflow uniformity (Gini coefficient), and **(h)** baseflow index

**Table A1.** Summary of streamflow and catchment mean meteorological time-series for a period from 01-01-1980 to 31-12-2020.

| Time series class | Variable name | Description | Unit | Data source / Reference |
|---|---|---|---|---|
| Hydrological time series | streamflow_observed | available observed streamflow time series | m³/s | IndiaWRIS |
| | LSTM_pred_streamflow | regionally trained LSTM-based hydrological model predicted streamflow time series | m³/s | (Mangukiya et al., 2023) |
| Meteorological time series | prcp | precipitation | mm/day | IMD (Pai et al., 2014) |
| | tmax | maximum temperature | °C | IMD (Srivastava et al., 2009) |
| | tmin | minimum temperature | °C | |
| | tavg | averaged temperature | °C | (tmax + tmin) / 2 |
| | srad_lw | surface downward long-wave radiation flux | w/m² | IMDAA (Rani et al., 2021) |





| | | | |
|---|---|---|---|
| srad_sw | surface downward short-wave radiation flux | w/m$^2$ | |
| wind_u | U-component of wind (10 m) | m/s | |
| wind_v | V-component of wind (10 m) | m/s | |
| wind | averaged wind speed (10 m) | m/s | $\sqrt{(\text{wind\_u}^2 + \text{wind\_v}^2)}$ |
| rel_hum | relative humidity (2 m) | % | |
| evap_canopy | evaporation rate from canopy | mm/day | IMDAA (Rani et al., 2021) |
| evap_surface | evaporation rate from the soil surface | mm/day | |
| pet | potential evapotranspiration (1981-2020) | mm/day | (Singer et al., 2021) |
| pet_gleam | potential evapotranspiration | mm/day | GLEAM (Miralles et al., 2011) |
| aet_gleam | actual evapotranspiration | mm/day | |
| sm_lvl1 | soil moisture of layer 1 (0-0.1 m below ground) | kg/m$^2$ | |
| sm_lvl2 | soil moisture of layer 2 (0.1-0.35 m below ground) | kg/m$^2$ | IMDAA (Rani et al., 2021) |
| sm_lvl3 | soil moisture of layer 3 (0.35-1 m below ground) | kg/m$^2$ | |
| sm_lvl4 | soil moisture of layer 4 (1-3 m below ground) | kg/m$^2$ | |


**Table A2.** Summary of catchment attributes representing gauge names and identifiers.

| Attribute | Description | Unit | Data source / Reference |
|---|---|---|---|
| gauge_id | gauge station identifier (5-digit; first 2 digits are cwc basin code and last 3 digits are station number) | - | |
| ghi_stn_id | unique ID used to identify a station, 10 characters long | - | GHI (Goteti, 2023) |
| cwc_site_name | name of the station | - | CWC |
| river_basin | name of the river basin | - | CWC |
| cwc_river | river/tributary | - | CWC |
| ghi_group | ghi assigned group (G1 or G2) | - | GHI (Goteti, 2023) |
| flow_availability | percentage duration for which streamflow data is available between 1980-2020 | % | CWC |

**Table A3.** Summary of catchment attributes representing location and topography.

| Attribute | Description | Unit | Data source / Reference |
|---|---|---|---|
| cwc_lat | latitude of the station | decimal degrees | CWC |





| cwc_lon | longitude of the station | decimal degrees | CWC |
| ghi_lat | latitude of the ghi relocated station | decimal degrees | GHI (Goteti, 2023) |
| ghi_lon | longitude of the ghi relocated station | decimal degrees | GHI (Goteti, 2023) |
| elev_mean | catchment mean elevation | m | SRTM DEM 90m |
| elev_median | catchment median elevation | m | SRTM DEM 90m |
| elev_min | catchment min elevation | m | SRTM DEM 90m |
| elev_max | catchment max elevation | m | SRTM DEM 90m |
| slope_mean | catchment mean slope | % | SRTM DEM 90m |
| slope_median | catchment median slope | % | SRTM DEM 90m |
| slope_min | catchment min slope | % | SRTM DEM 90m |
| slope_max | catchment max slope | % | SRTM DEM 90m |
| cwc_area | catchment drainage area | km$^2$ | CWC |
| ghi_area | catchment drainage area | km$^2$ | GHI (Goteti, 2023) |
| gauge_elevation | elevation of the gauging station | m | SRTM DEM 90m |
| dpsbar | catchment mean drainage path slope | m/km | SRTM DEM 90m |

**Table A4.** Summary of catchment attributes representing climate indices computed from 01-01-1980 to 31-12-2020.

| Attributes | Description | Unit | Data Source / Reference |
|---|---|---|---|
| p_mean | mean daily precipitation | mm/day | IMD |
| p_max | maximum daily precipitation | mm/day | IMD |
| p_mean_anum | annual average total precipitation | mm | IMD |
| p_monthly_variability | variation in precipitation patterns throughout the year (higher values indicate greater variation) | - | IMD |
| p_annual_variability | variation in annual precipitation patterns (higher values indicate greater variation) | - | IMD |
| p_unif | how uniformly the precipitation is distributed in a year, 0 if the annual maximum precipitation is uniformly distributed throughout the year, 1 if the annual maximum precipitation occurs in a single day | - | IMD |
| high_prec_freq | frequency of high precipitation days ($\geq 5$ times the mean daily precipitation) | days/year | IMD |
| high_prec_dur | average duration of high precipitation events (number of consecutive days $\geq 5$ times the mean daily precipitation) | days | IMD |
| max_high_prec_dur | maximum number of consecutive days with precipitation $\geq 5$ times the mean daily precipitation | days | IMD |
| high_prec_timing | season during which most high precipitation days ($\geq 5$ times the mean daily precipitation) occur | season | IMD |
| low_prec_freq | frequency of dry days (precipitation < 1 mm/day) | days/year | IMD |





| | | | |
|---|---|---|---|
| low_prec_dur | average duration of dry periods (number of consecutive days < 1 mm/day) | days | IMD |
| max_low_prec_dur | maximum number of consecutive days with precipitation < 1 mm/day | days | IMD |
| low_prec_timing | Season during which most dry days (< 1 mm/day) occur | season | IMD |
| asynchronicity | asynchronicity between the annual precipitation and PET cycles, where high values represent high relative magnitude and phase differences | - | (Feng et al., 2019) |
| tmin_mean | mean daily minimum temperature | °C | IMD |
| tmax_mean | mean daily maximum temperature | °C | IMD |
| pet_mean | mean daily potential evapotranspiration | mm/day | (Singer et al., 2021) |
| pet_min | minimum daily potential evapotranspiration | mm/day | (Singer et al., 2021) |
| pet_max | maximum daily potential evapotranspiration | mm/day | (Singer et al., 2021) |
| pet_mean_anum | annual average total potential evapotranspiration | mm | (Singer et al., 2021) |
| pet_gleam_mean | mean daily average potential evapotranspiration | mm/day | (Miralles et al., 2011) |
| aet_gleam_mean | mean daily average actual evapotranspiration | mm/day | (Miralles et al., 2011) |
| evap_canopy_mean | mean daily evaporation rate from the canopy | mm/day | IMDAA |
| evap_canopy_min | minimum daily evaporation rate from the canopy | mm/day | IMDAA |
| evap_canopy_max | maximum daily evaporation rate from the canopy | mm/day | IMDAA |
| evap_canopy_anum | annual average total evaporation from the canopy | mm | IMDAA |
| evap_surface_mean | mean daily evaporation rate from the soil surface | mm/day | IMDAA |
| evap_surface_min | minimum daily evaporation rate from the soil surface | mm/day | IMDAA |
| evap_surface_max | maximum daily evaporation rate from the soil surface | mm/day | IMDAA |
| evap_surface_anum | annual average total evaporation from the soil surface | mm | IMDAA |
| aridity_p_pet | aridity index (P/PET; ratio of mean annual precipitation over the mean annual potential evapotranspiration) | - | |
| aridity_pet_aet | aridity index [(PET-AET)/PET; a ratio of the deficit between potential and actual evapotranspiration over potential evapotranspiration] | - | |
| ai_mean | spatially averaged aridity index of the catchment | - | (Trabucco and Zomer, 2018) |
| rel_hum_mean | mean daily relative humidity (2 m) | % | IMDAA |
| srad_lw_mean | mean daily surface downward long-wave radiation flux | w/m$^2$ | IMDAA |
| srad_sw_mean | mean daily surface downward short-wave radiation flux | w/m$^2$ | IMDAA |
| wind_mean | mean daily wind speed (10 m) | m/s | IMDAA |
| sm_lvl1_mean | mean daily soil moisture in layer 1 (0-0.1 m below ground) | kg/m$^2$ | IMDAA |



| | | | |
|---|---|---|---|
| sm_lvl2_mean | mean daily soil moisture in layer 2 (0.1-0.35 m below ground) | kg/m$^2$ | IMDAA |
| sm_lvl3_mean | mean daily soil moisture in layer 3 (0.35-1 m below ground) | kg/m$^2$ | IMDAA |
| sm_lvl4_mean | mean daily soil moisture in layer 4 (1-3 m below ground) | kg/m$^2$ | IMDAA |

**Table A5.** Summary of catchment attributes representing hydrological signatures computed for 01-01-1980 to 31-12-2015.

| Attributes | Description | Unit | Data Source / Reference |
|---|---|---|---|
| q_mean | mean daily streamflow of the catchment | mm/day | IndiaWRIS |
| runoff_ratio | runoff ratio (ratio of mean daily streamflow to the mean daily precipitation of catchment) | - | |
| streamflow_elas | streamflow precipitation elasticity (i.e., the sensitivity of streamflow to changes in precipitation at the annual timescale, using the mean daily discharge as reference) | - | Eq. (7) in (Sankarasubramanian et al., 2001) |
| slope_fdc | slope of the flow duration curve between the log-transformed 33rd and 66th streamflow percentiles | - | (Addor et al., 2017) |
| bfi | baseflow index, computed as the ratio of mean daily baseflow to mean daily discharge, with the hydrograph separation performed using the Ladson et al. (2013) digital filter | - | |
| q_cv | variability of daily streamflow values (coefficient of variation) | % | IndiaWRIS |
| q_10 | first decile of mean daily streamflow (the value below which 10% of the observations fall) | mm/day | IndiaWRIS |
| q_25 | first quartile of mean daily streamflow (the value below which 25% of the observations fall) | mm/day | IndiaWRIS |
| q_50 | median of mean daily streamflow (the value below which 50% of the observations fall) | mm/day | IndiaWRIS |
| q_75 | third quartile of mean daily streamflow (the value below which 75% of the observations fall) | mm/day | IndiaWRIS |
| q_90 | 90th percentile of mean daily streamflow (the value below which 90% of the observations fall; High flows) | mm/day | IndiaWRIS |
| q_zero | frequency of days with zero flow | days/year | IndiaWRIS |
| q_low_days | mean number of consecutive days with flow less than 25th percentile mean daily flow | days | IndiaWRIS |
| freq_q_low | frequency of days with low flows (flow less than 25th percentile mean daily flow) | days/year | IndiaWRIS |
| q_high_days | mean number of consecutive days with a flow more than the 95th percentile mean daily flow | days | IndiaWRIS |
| freq_q_high | frequency of days with high flows (flow more than 95th percentile mean daily flow) | days/year | IndiaWRIS |
| annual_q | mean annual flow volume in the catchment | MCM/year | IndiaWRIS |
| mean_anum_flow | mean annual flow volume in the catchment (computed for 1950 to 2020) | MCM/year | GHI (Goteti, 2023) |
| cen_time | centre timing, corresponds to day of the year (doy) at which 50% of annual flow is reached | Day | |





| | | | |
|---|---|---|---|
| gini_flow | uniformity of flow over the days in a year; 0 indicates equal flow throughout the year, and 1 indicates all flow occurred in a single day | - | |
| annual_max_1day | mean annual 1-day maximum flow | m³/s | IndiaWRIS |
| annual_max_3day | mean annual 3-day maximum flow | m³/s | IndiaWRIS |
| annual_max_7day | mean annual 7-day maximum flow | m³/s | IndiaWRIS |
| annual_max_30day | mean annual 30-day maximum flow | m³/s | IndiaWRIS |
| annual_max_90day | mean annual 90-day maximum flow | m³/s | IndiaWRIS |
| annual_min_7day | mean annual 7-day minimum flow | m³/s | IndiaWRIS |
| month_1day_max | month of 1-day maximum flow for the majority of the years | calendar month | IndiaWRIS |
| month_1day_min | month of 1-day minimum flow for the majority of the years | calendar month | IndiaWRIS |
| doy_min_flow | the day of the year (doy) at which minimum streamflow occurred | Day | |
| doy_max_flow | the day of the year (doy) at which maximum streamflow occurred | Day | |
| doy_min_flow_7 | the day of the year (doy) at which minimum 7-day streamflow occurred | Day | |
| doy_max_flow_7 | the day of the year (doy) at which maximum 7-day streamflow occurred | Day | |
| mean_jan_flow | mean monthly flow volume of January in the catchment (computed for 1950 to 2020) | MCM/month | GHI (Goteti, 2023) |
| mean_feb_flow | mean monthly flow volume of February in the catchment (computed for 1950 to 2020) | MCM/month | GHI (Goteti, 2023) |
| mean_mar_flow | mean monthly flow volume of March in the catchment (computed for 1950 to 2020) | MCM/month | GHI (Goteti, 2023) |
| mean_apr_flow | mean monthly flow volume of April in the catchment (computed for 1950 to 2020) | MCM/month | GHI (Goteti, 2023) |
| mean_may_flow | mean monthly flow volume of May in the catchment (computed for 1950 to 2020) | MCM/month | GHI (Goteti, 2023) |
| mean_jun_flow | mean monthly flow volume of June in the catchment (computed for 1950 to 2020) | MCM/month | GHI (Goteti, 2023) |
| mean_jul_flow | mean monthly flow volume of July in the catchment (computed for 1950 to 2020) | MCM/month | GHI (Goteti, 2023) |
| mean_aug_flow | mean monthly flow volume of August in the catchment (computed for 1950 to 2020) | MCM/month | GHI (Goteti, 2023) |
| mean_sep_flow | mean monthly flow volume of September in the catchment (computed for 1950 to 2020) | MCM/month | GHI (Goteti, 2023) |
| mean_oct_flow | mean monthly flow volume of October in the catchment (computed for 1950 to 2020) | MCM/month | GHI (Goteti, 2023) |
| mean_nov_flow | mean monthly flow volume of November in the catchment (computed for 1950 to 2020) | MCM/month | GHI (Goteti, 2023) |
| mean_dec_flow | mean monthly flow volume of December in the catchment (computed for 1950 to 2020) | MCM/month | GHI (Goteti, 2023) |
| cv_jan_flow | variability of daily streamflow values in January | % | IndiaWRIS |
| cv_feb_flow | variability of daily streamflow values in February | % | IndiaWRIS |



| | | | |
|---|---|---|---|
| cv_mar_flow | variability of daily streamflow values in March | % | IndiaWRIS |
| cv_apr_flow | variability of daily streamflow values in April | % | IndiaWRIS |
| cv_may_flow | variability of daily streamflow values in May | % | IndiaWRIS |
| cv_jun_flow | variability of daily streamflow values in June | % | IndiaWRIS |
| cv_jul_flow | variability of daily streamflow values in July | % | IndiaWRIS |
| cv_aug_flow | variability of daily streamflow values in August | % | IndiaWRIS |
| cv_sep_flow | variability of daily streamflow values in September | % | IndiaWRIS |
| cv_oct_flow | variability of daily streamflow values in October | % | IndiaWRIS |
| cv_nov_flow | variability of daily streamflow values in November | % | IndiaWRIS |
| cv_dec_flow | variability of daily streamflow values in December | % | IndiaWRIS |
| mean_swmn_flow | mean flow volume of the southwest monsoon season (June, July, Aug, Sept) in the catchment (computed for 1950 to 2020) | MCM/season | GHI (Goteti, 2023) |
| mean_atmn_flow | mean flow volume of autumn/retreating monsoon season (Oct, Nov) in the catchment (computed for 1950 to 2020) | MCM/season | GHI (Goteti, 2023) |
| mean_wint_flow | mean flow volume of the winter season (Dec, Jan, Feb) in the catchment (computed for 1950 to 2020) | MCM/season | GHI (Goteti, 2023) |
| mean_sumr_flow | mean flow volume of the summer season (Mar, Apr, May) in the catchment (computed for 1950 to 2020) | MCM/season | GHI (Goteti, 2023) |
| q_mean_swmn | mean daily streamflow of the southwest monsoon season (June, July, Aug, Sept) in a catchment | mm/day | IndiaWRIS |
| q_5_swmn | 5th percentile of daily streamflow in southwest monsoon season (June, July, Aug, Sept) | mm/day | IndiaWRIS |
| q_25_swmn | first quartile of daily streamflow in southwest monsoon season (June, July, Aug, Sept) | mm/day | IndiaWRIS |
| q_50_swmn | median of daily streamflow in southwest monsoon season (June, July, Aug, Sept) | mm/day | IndiaWRIS |
| q_75_swmn | third quartile of daily streamflow in southwest monsoon season (June, July, Aug, Sept) | mm/day | IndiaWRIS |
| q_95_swmn | 95th percentile of daily streamflow in southwest monsoon season (June, July, Aug, Sept) | mm/day | IndiaWRIS |
| rise_rate_mean | mean of all positive differences between consecutive daily flows | $m^3/s$ | IndiaWRIS |
| rise_rate_median | median of all positive differences between consecutive daily flows | $m^3/s$ | IndiaWRIS |
| rise_days | mean number of days in a year with positive differences between consecutive daily flows | days/year | IndiaWRIS |
| fall_rate_mean | mean of all negative differences between consecutive daily flows | $m^3/s$ | IndiaWRIS |
| fall_rate_median | median of all negative differences between consecutive daily flows | $m^3/s$ | IndiaWRIS |
| fall_days | mean number of days in a year with negative differences between consecutive daily flows | days/year | IndiaWRIS |
| num_hyd_alt | mean number of hydrologic reversals in a year (change from rise to fall) | - | IndiaWRIS |



**Table A6.** Summary of catchment attributes representing land cover characteristics.

| Attributes | Description | Unit | Data Source / Reference |
|---|---|---|---|
| water_frac | water cover fraction (2017 - 2022) | - | ESRI land cover (Karra et al., 2021) |
| trees_frac | trees cover fraction (2017 - 2022) | - | ESRI land cover |
| flooded_veg_frac | flooded vegetation fraction (2017 - 2022) | - | ESRI land cover |
| crops_frac | crop cover fraction (2017 - 2022) | - | ESRI land cover |
| built_area_frac | urban cover fraction (2017 - 2022) | - | ESRI land cover |
| bare_frac | bare cover fraction (2017 - 2022) | - | ESRI land cover |
| range_frac | range cover fraction (2017 - 2022) | - | ESRI land cover |
| dom_land_cover | dom_land cover type (2017 - 2022) | - | ESRI land cover |
| dom_land_cover_frac | dom_land cover fraction (2017 - 2022) | - | ESRI land cover |
| lai_mean | catchment mean leaf area index (2001-2020) | - | |
| lai_min | minimum leaf area index (2001-2020) | - | MODIS MCD15A2H (Myneni et al., 2015) |
| lai_max | maximum leaf area index (2001 - 2020) | - | |
| lai_diff | difference between maximum and minimum leaf area index (2001 - 2020) | - | |


**Table A7.** Summary of catchment attributes representing soil characteristics.

| Attributes | Description | Unit | Data Source / Reference |
|---|---|---|---|
| soil_depth | mean soil and sedimentary-deposit thickness | m | (Pelletier et al., 2016) |
| soil_conductivity_top | mean saturated hydraulic conductivity of topsoil (30 - 200 cm) | cm/day | HiHydroSoil v2 (Simons et al., 2020) |
| soil_conductivity_sub | mean saturated hydraulic conductivity of subsoil (0 - 30 cm) | cm/day | HiHydroSoil v2 |
| soil_awc_top | mean available water content of topsoil (30 - 200 cm) | $m^3/m^3$ | HiHydroSoil v2 |
| soil_awc_sub | mean available water content of subsoil (0 - 30 cm) | $m^3/m^3$ | HiHydroSoil v2 |
| soil_awsc_min | minimum available water storage capacity of the soil | mm/m | FAO Soil Data |
| soil_awsc_max | maximum available water storage capacity of the soil | mm/m | FAO Soil Data |
| soil_awsc_major | available water storage capacity of the soil for the majority part of the catchment | mm/m | FAO Soil Data |
| sand_frac_top | fraction of sand in topsoil (0 - 30 cm) for the majority of the catchment area | %wt | HWSD v2 (FAO and IISA, 2023) |
| sand_frac_sub | fraction of sand in subsoil (30 - 100 cm) for the majority of the catchment area | %wt | HWSD v2 |
| silt_frac_top | fraction of silt in topsoil (0 - 30 cm) for the majority of the catchment area | %wt | HWSD v2 |
| silt_frac_sub | fraction of silt in subsoil (30 - 100 cm) for the majority of the catchment area | %wt | HWSD v2 |



| | | | |
|---|---|---|---|
| clay_frac_top | fraction of clay in topsoil (0 - 30 cm) for the majority of the catchment area | %wt | HWSD v2 |
| clay_frac_sub | fraction of clay in subsoil (30 - 100 cm) for the majority of the catchment area | %wt | HWSD v2 |
| gravel_frac_top | fraction of gravel in topsoil (0 - 30 cm) for the majority of the catchment area | %vol | HWSD v2 |
| gravel_frac_sub | fraction of gravel in subsoil (30 - 100 cm) for the majority of the catchment area | %vol | HWSD v2 |
| bulkdens_top_major | reference bulk density of topsoil (0 - 30 cm) for the majority of the catchment area | kg/dm$^3$ | HWSD v2 |
| bulkdens_top_mean | mean reference bulk density of topsoil (0 - 30 cm) | kg/dm$^3$ | HWSD v2 |
| bulkdens_sub_major | reference bulk density of subsoil (30 - 100 cm) for the majority of the catchment area | kg/dm$^3$ | HWSD v2 |
| bulkdens_sub_mean | mean reference bulk density of subsoil (30 - 100 cm) | kg/dm$^3$ | HWSD v2 |
| org_carb_top_major | organic carbon content in topsoil (0 - 30 cm) for the majority of the catchment area | %wt | HWSD v2 |
| org_carb_top_mean | mean organic carbon content in topsoil (0 - 30 cm) | %wt | HWSD v2 |
| org_carb_sub_major | organic carbon content in subsoil (30 - 100 cm) for the majority of the catchment area | %wt | HWSD v2 |
| org_carb_sub_mean | mean organic carbon content in subsoil (30 - 100 cm) | %wt | HWSD v2 |
| organic_frac_top | mean fraction of organic matter content in topsoil (30 - 200 cm) | - | HiHydroSoil v2 (Simons et al., 2020) |
| organic_frac_sub | mean fraction of organic matter content in subsoil (0 - 30 cm) | - | HiHydroSoil v2 |
| hsg_major | hydrological soil group for the majority of the catchment area | - | HiHydroSoil v2 |
| wtd | catchment mean water table depth | m | (Fan et al., 2013) |

**Table A8.** Summary of catchment attributes representing geological characteristics.

| Attributes | Description | Unit | Data Source / Reference |
|---|---|---|---|
| geol_porosity | mean subsurface porosity | - | GLHYMPS (Gleeson et al., 2014) |
| geol_permeability | mean subsurface permeability | m$^2$ | GLHYMPS |
| geol_class_1st | most common geological class in a catchment | - | GLiM (Hartmann and Moosdorf, 2012) |
| geol_class_1st_frac | fraction of catchment area associated with its most common geological class | - | GLiM |
| geol_class_2nd | second most common geological class in the catchment | - | GLiM |
| geol_class_2nd_frac | fraction of catchment area associated with its second most common geological class | - | GLiM |
| carb_rocks_frac | fraction of catchment area characterized as "carbonated sedimentary rocks" | - | GLiM |

**Table A9.** Summary of catchment attributes representing anthropogenic influences.



| Attributes | Description | Unit | Data Source / Reference |
|---|---|---|---|
| num_dams | total number of large and medium dams in catchments | - | IndiaWRIS |
| res_store_sum | sum of total volume content of dams within the catchment | $10^3$ m$^3$ | IndiaWRIS |
| n_dams | total number of dams in a catchment | - | GRaND (Lehner et al., 2011) |
| first_dam_year | year of construction of the first dam | - | GRaND |
| latest_dam_year | year of construction of the recent dam | - | GRaND |
| total_storage | total storage of the reservoirs | m$^3$ | GRaND |
| reservoir_index | ratio of total storage to multi-year annual streamflow | - | GRaND |
| irrigation_frac | percentage of dams used for irrigation | - | GRaND |
| hydroelec_frac | percentage of dams used for hydroelectric generation | - | GRaND |
| drinking_frac | percentage of dams used for drinking | - | GRaND |
| flood_frac | percentage of dams used for flood storage | - | GRaND |
| overflow_frac | percentage of dams used for overflow control | - | GRaND |
| navigation_frac | percentage of dams used for navigation | - | GRaND |
| tailings_frac | percentage of dams used for tailings (storing by products of mining operations) | - | GRaND |
| pop_density_2000 | averaged population density of the catchment in 2000 | people/km$^2$ | data.humdata.org (WorldPop and CIESIN, 2018) |
| pop_density_2005 | averaged population density of the catchment in 2005 | people/km$^2$ | data.humdata.org |
| pop_density_2010 | averaged population density of the catchment in 2010 | people/km$^2$ | data.humdata.org |
| pop_density_2015 | averaged population density of the catchment in 2015 | people/km$^2$ | data.humdata.org |
| pop_density_2020 | averaged population density of the catchment in 2020 | people/km$^2$ | data.humdata.org |
| urban_frac_1985 | fraction of urban land cover in a catchment in 1985 | - | (Roy et al., 2015) |
| urban_frac_1995 | fraction of urban land cover in a catchment in 1995 | - | (Roy et al., 2015) |
| urban_frac_2005 | fraction of urban land cover in a catchment in 2005 | - | (Roy et al., 2015) |
| crops_frac_1985 | fraction of cropland land cover in a catchment in 1985 | - | (Roy et al., 2015) |
| crops_frac_1995 | fraction of cropland land cover in a catchment in 1995 | - | (Roy et al., 2015) |
| crops_frac_2005 | fraction of cropland land cover in a catchment in 2005 | - | (Roy et al., 2015) |

**Author contribution**

**Nikunj K. Mangukiya:** Conceptualization; Data curation; Formal analysis; Investigation; Visualization; Writing-original draft preparation. **Kanneganti Bhargav Kumar:** Conceptualization; Data curation; Formal analysis; Investigation; Writing-original draft preparation. **Pankaj Dey:** Conceptualization; Writing-review & editing. **Shailza Sharma:** Conceptualization;

Writing-review & editing. **Vijaykumar Bejagam:** Data curation; Formal analysis. **P. P. Mujumdar:** Conceptualization; Resources; Supervision; Writing-review & editing. **Ashutosh Sharma:** Conceptualization; Resources; Supervision; Writing-review & editing.

**Competing interests:**

The authors declare that they have no conflict of interest.

**Acknowledgments**

The authors gratefully acknowledge the Central Water Commission (CWC), the National Water Informatics Centre (NWIC), and the Ministry of Jal Shakti (MoJS) for providing the streamflow dataset through the online portal, India – Water Resources Information System (India-WRIS; https://indiawris.gov.in/wris/#/). The authors also extend their gratitude to the India Meteorological Department (IMD), Ministry of Earth Sciences, Government of India, for providing the gridded rainfall and 570 temperature datasets through their respective websites. Additionally, the authors gratefully acknowledge the National Centre for Medium Range Weather Forecasting (NCMRWF), Ministry of Earth Sciences, Government of India, for the Indian Monsoon Data Assimilation and Analysis (IMDAA) reanalysis. The IMDAA reanalysis was produced under the collaboration between UK Met Office, NCMRWF, and IMD, with financial support from the Ministry of Earth Sciences under the National Monsoon Mission programme. The authors utilized numerous publicly available datasets for compiling catchment attributes 575 and meteorological forcing time series, duly acknowledging and citing them where applicable. The authors extend their gratitude to all the researchers and contributing authors of these open-source datasets.

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
