# Peer review of "CAMELS-IND: hydrometeorological time series and catchment attributes for 228 catchments in Peninsular India"

_Earth System Science Data, 2024_

## Author Comment (AC1)

Dear Ashish Manoj J.,

Thank you for your efforts in reviewing our manuscript. We are extremely grateful to you for your thoughtful recommendations and questions on methodology. We provide here our responses to your comments and mention the changes made in the manuscript.

Major Comments:

1) ESSD generally encourages the sharing of all relevant processing steps and code required to replicate the results (Carlson and Oda, 2018). This is particularly important for datasets to build user confidence and to adhere to FAIR principle. A flowchart could be added to the Appendix detailing the different products and steps used in generating the dataset. Another possible suggestion is to create a separate repository to deposit all the relevant codes and link them to the data availability statement. Similar processing pipelines are already established for the CAMELS-DE (Dolich, 2024) and CAMELS-CH (https://camels-ch.github.io/).

**Response:** Thank you for your suggestion. All the processing steps have been described in the manuscript [Line 193-194, 219-220, 224, 332, 361-362, 381-382]. The data products used to derive the meteorological time series and catchment attributes are already provided in Table A1 to A9. The meteorological time series were derived from gridded data products using area-weighted averages at daily scale. Most of the catchment attributes, including topography, land use, soil, geology, and anthropogenic characteristics were derived using zonal statistics and zonal histogram tool of QGIS or directly compiled from the source data at catchment scale. We plan to create a separate repository in future to include relevant scripts for deriving climate indices and hydrological signatures, with a guiding document. This repository will then be linked to the data repository.

2) I went through the Zenodo entry and found that the dataset was previously named CAMELS-IND rather than CAMELS-INDIA. I feel that the former version better aligns with the naming conventions of other CAMELS products. In spirit of ESSD open discussion and for the benefit of future readers, I would like to raise this point so that the authors can reply with their reasoning here.

**Response:** Thank you for raising this point. Based on yours and other reviewer suggestion, we have revised the name of the dataset as "CAMELS-IND" to better align with the naming conventions of other CAMELS datasets.

3) Attribute file naming: This is again linked to my previous comment. Generally, only small letters are preferred in file names as this would aid in automation of code pipelines and other scripts. Hence for example, I would suggest camels_ind_name or camels_india_name instead of.

**Response:** Thank you for your feedback. Since modern scripting languages and tools typically handle file names in both uppercase and lowercase without issues, the use of capital letters in file naming should not impact usability or automation process. However, we renamed all files with small letters as "camels_ind_xxxx" to enhance consistency and user experience across different systems [Table 1, Line 206].

4) I have a minor concern regarding the different zip files for each folder. In general, this makes it more tedious to download and extract each file individually. The total file size seems to be under 1 GB only in any case, it would be worthwhile to consider having a single zip file with subfolders for the entire dataset (similar to the Caravan file structure).

**Response:** Thank you for your suggestions. The dataset is now provided as a single ZIP file containing subfolders for catchment mean forcings, attributes, shapefiles, and streamflow data, along with a changelog, disclaimer, and data description file.

5) I would also recommend adding the license/disclaimer as a text file within the dataset to ensure this is readily available when a user directly downloads the product.

**Response:** The disclaimer has now been added as a text file within the dataset.

6) Some vital information is missing in Section 7. Consider adding more details (including dataset access DOI) about the specific GLDAS model (Noah/CLM/VIC) and versioning (with

or without GRACE- Data Assimilation) used for the preliminary quality assessment. The same can also be added as a dataset citation.

**Response:** Thank you for your valuable suggestion. The Global Land Data Assimilation System (GLDAS) data used for preliminary quality assessment was obtained from the GLDAS Noah Land Surface Model L4, with 3-hourly data at 0.25° x 0.25° resolution. Version 2.0 was used for the period 1980–2014 (Beaudoing and Rodell, 2019), and version 2.1 for 2014–2020 (Beaudoing and Rodell, 2020). The 3-hourly data was then resampled to daily data and used for the preliminary quality assessment. Details and dataset citation have been added to the manuscript [Line 463-465].

Beaudoing, H. and M. Rodell, NASA/GSFC/HSL (2019), GLDAS Noah Land Surface Model L4 3 hourly 0.25 x 0.25 degree V2.0, Greenbelt, Maryland, USA, Goddard Earth Sciences Data and Information Services Center (GES DISC), Accessed: [22 Nov 2022]. https://doi.org/10.5067/342OHQM9AK6Q

Beaudoing, H. and M. Rodell, NASA/GSFC/HSL (2020), GLDAS Noah Land Surface Model L4 3 hourly 0.25 x 0.25 degree V2.1, Greenbelt, Maryland, USA, Goddard Earth Sciences Data and Information Services Center (GES DISC), Accessed: [26 Nov 2022]. https://doi.org/10.5067/E7TYRXPJKWOQ

7) In the catchment and station shapefiles, I could find some minor mismatches between the flow outlets and catchment boundaries (For example, Station – 15028 (Thiruvattar)). I think this has already been mentioned in Goteti (2023) and the manuscript. I would again mention this in Section 8 so that future users are aware of possible mismatch issues.

**Response:** Thank you for your concern. We have now addressed this limitation in the revised manuscript as follows: "*The catchment boundaries included in the CAMELS-IND dataset are derived from Goteti (2023), who highlighted that the catchment delineation based on the 500 m HydroSHEDS v1 dataset may suffice for larger catchments but could introduce minor mismatches, particularly in smaller catchments and around flow outlets. The upcoming HydroSHEDS v2, with 12 m topographic data, is anticipated to improve spatial accuracy, enhancing the delineation of catchment boundaries and river networks for future dataset versions.*" [Line 520-524]

Minor Comments:

I have left a few minor comments on the annotated version of the manuscript. Some are more subjective and personal than others. Feel free to make changes that you feel fit.

Line 64: I am not sure if Caravan could be considered as a cloud based platform. While it does provide an interface to Google Earth Engine Cloud services. I would opt for more apt wording here. Possible suggestions - community-driven initiative

**Response:** Thank you for your comment. We have revised the sentence as follows: "*The community-driven initiative further extended Caravan datasets for Denmark (Koch, 2022) and Israel (Efrat, 2023).*" [Line 65-66]

Line 82: Since Goteti (2023) is already cited earlier in this paragraph and the next section details this work. I would suggest adding other relevant references here.

**Response:** Thank you for the suggestion. We reviewed the literature but found that Goteti (2023) remains the most relevant and comprehensive source for this context. To our knowledge, no additional studies address this issue with similar detail, so we have retained this citation.

Line 120: I liked this paragraph and the motivation of using the better quality national products. To further strengthen this argument you could add that recent studies have uncovered deficiencies in globally available products such as ERA5 Land (potential evapotranspiration : F. Clerc-Schwarzenbach et al. 2024) compared to the national products used in CAMELS.

Maria Clerc-Schwarzenbach, F., Selleri, G., Neri, M., Toth, E., van Meerveld, I., Seibert, J., 2024. HESS Opinions: A few camels or a whole caravan? Hydrol. Earth Syst. Sci. 1–29.

**Response:** Thank you for your positive feedback. While we appreciate the suggestion to include recent studies that identify deficiencies in globally available products compared to national data sources, specifically for the US, Brazil, and Great Britain, we believe that our preliminary assessments already provide sufficient evidence of the advantages of the national datasets utilized in CAMELS-IND for India. [Line 486-488]

Line 147: I would give the link to the start site for WRIS (https://indiawris.gov.in) as the current link seems broken to me. Date of Access - 08.10.2024

**Response:** Thank you for the suggestion. The links are updated throughout the manuscript as "https://indiawris.gov.in/wris/" [Line 70, 150, 516]

Line 181-182: Add citation to CWC Atlas or other sources if possible.

**Response:** Thank you. The citation has been added as "India-WRIS (2012)". [Line 200]

India-WRIS: River Basin Atlas of India, RRSC-West, NRSC, ISRO, Jodhpur, India, 1–144 pp., 2012. https://indiawris.gov.in/wris/#/atlas

Line 89: Check for typo; Line 91-92: typo?; Line 96: provided?; Line 134: Palghat Gap (Palakkad Gap); Line 136: Bharathapuzha; Line 154: 2020?; Line 160: 2020?; Line 167: u and v components; Line 237: south eastern part?; Line 249: wind speed

**Response:** Thank you. We have corrected all the typos and incorporated all suggestions in the manuscript.

Line 315: Citation?

**Response:** Thank you. We have added the citation as "Myneni et al., 2015". [Line 334]

Myneni, R., Knyazikhin, Y., and Park, T.: MCD15A2H MODIS/Terra+Aqua Leaf Area Index/FPAR 8-day L4 Global 500m SIN Grid V006 [Data set], NASA EOSDIS L. Process. DAAC, https://doi.org/10.5067/MODIS/MCD15A2H.006, 2015.

Line 449-450: Is this the GLDAS - 2.1 Model? Please add more information and relevant dataset citation here.

**Response:** Thank you. As stated earlier, we used GLDAS Noah Land Surface Model L4 3 hourly 0.25 x 0.25-degree V2.0 and V2.1 for preliminary assessment of dataset. The data citation has been added. [Line 463-465]

Line 463-464: Is the testing and training period same as before?

**Response:** Thank you for raising this question. We have added a statement to clarify that "*The model was trained on data from 1991 to 2011 and tested on data from 2011 to 2015.*" [Line 480]

Line 532: Add website citation/acknowledgment.

**Response:** Thank you for the suggestion. We have added the image credits in the caption for Figure A1 and A2.

Overall, I feel the manuscript could have a moderate revision before it can finally be accepted in ESSD.

We greatly appreciate your feedback. We believe these changes address the concerns raised and improve the quality and clarity of the manuscript. Thank you once again.

Best regards,

Ashutosh Sharma (on behalf of all co-authors)

---

## Author Comment (AC2)

Dear Dr. Gemma Coxon,

We sincerely thank you for your time and efforts in reviewing our manuscript and offering constructive remarks to improve the manuscript. We provide here our responses to your comments and concerns and mention the changes made in the manuscript (marked with *Italic* text).

This paper describes the CAMELS-INDIA dataset, which consists of hydrometeorological time series and catchment attributes for a large sample of catchments in India. This will be a valuable dataset for the hydrological community. Overall, the paper is well-written, figures are well-produced and dataset well-described.

**Response:** We appreciate your feedback.

My major comment is the lack of observed streamflow data for many of the catchments. A key characteristic of CAMELS datasets is providing observed streamflow data – without this, the datasets are less useful for large-sample hydrological analyses. I was surprised that from the 472 catchments included in the dataset nearly a third (32%) of the catchments have no observed streamflow timeseries, and 45% of the catchments have less than 10% of streamflow for the chosen timeperiod. Why is this? Why were these catchments included in the dataset? How do the authors envisage their use in large-sample hydrological analyses without streamflow? I appreciate that there are modelled flow timeseries, but the model performance for the other gauging stations was not particularly convincing that these would provide a good representation of streamflow in catchments with no data. My recommendation would be to only include catchments where you have good streamflow data i.e. where you have calculated hydrological signatures (228 catchments – still a very valuable dataset). If the authors disagree with this recommendation then much better justification and clarity needs to be added into the paper highlighting this limitation with the dataset (see additional comments below).

**Response:** Thank you for your insightful comments. Based on your recommendation, we have revised the Title to specify only the 228 catchments where streamflow data is available for at least 30% of the period from 1980 to 2020, aligning with other CAMELS datasets that focus on catchments with long-term streamflow data in the Title. Necessary revisions have been made throughout the manuscript, including abstract as "*We introduce CAMELS-IND*

*(Catchment Attributes and MEteorology for Large-sample Studies – India), a dataset containing hydrometeorological time series and catchment attributes for 472 catchments in Peninsular India, of which 228 catchments have observed streamflow data available for over 30% of the period between 1980 to 2020.*" [Line 12-13, 159-160]

"*In addition to these 228 catchments, there are 60 catchments with partial streamflow records, ranging from 2.4% (approximately one year) to 30% (about 12 years) of the period. Although these may not meet the threshold for extensive hydrological analysis, they offer valuable opportunities for pseudo-ungauged experiments and model testing or validation in data-limited environments.*" [Line 160-163]. To facilitate efficient selection of catchments based on users' analytical needs, we have included a "filter_catchment.py" script in the dataset to help users create specific subsets of the data.

As noted in the manuscript "*The observed streamflow data was compiled from the India-WRIS portal, which was launched in 2019. Since its launch, continuous efforts have been made to digitize the available data and update the information on the portal. We anticipate that, with time, observations from the rest of the gauges will be made available for users to download. Therefore, we extracted catchment mean meteorological forcings and static attributes for all 472 catchments. To facilitate immediate use by those specifically requiring catchments with streamflow observations, we have also provided a subset of the dataset with the 228 catchments within CAMELS-IND.*" [Line 164-168]

Despite the limited streamflow data for some catchments, we included the full set of 472 catchments in the dataset to support a variety of hydrological analyses, such as meteorological drought assessments, seasonal and spatial pattern/trend analyses of meteorological variables, and other studies that may not rely on streamflow records. We believe that including all catchments, alongside the subset of 228 catchments and the "filter_catchment.py" script, offers users flexibility in catchment selection and supports diverse analyses suited to their research objectives.

Aside from the major comment above, I only have minor/moderate comments for the authors to consider before publication:

1) I suggest to follow similar naming conventions to other CAMELS datasets and change the name of the dataset to CAMELS-IND and all files.

**Response:** Thank you for raising this point. We have revised the name of the dataset and all files as "CAMELS-IND" to better align with the naming conventions of other CAMELS datasets.

2) L50. 'to some relevant questions' is very vague – can you be more specific, or perhaps give one or two examples of 'relevant' questions?

**Response:** Thank you for your comment. We have revised the sentence to improve clarity as follows: "*The availability of such catchment datasets offers a new perspective to the research community, supporting solutions for key issues in water management, quantification and risk assessment of hydrologic extremes, understanding regional-scale hydrologic functioning, and assessing climate change impacts.*" [Line 50-52]

3) L60. I would not include CAMELS-FR in this list as it is not published yet (as far as I know) and this reference is an EGU conference abstract. The same for CAMELS-DK which is currently a pre-print and not published.

**Response:** Thank you for your suggestion. We have removed CAMELS-FR and CAMELS-DK from the reference list. Additionally, we have replaced CAMELS-ES with the recently published BULL database for Spain.

Senent-Aparicio, J., Castellanos-Osorio, G., Segura-Méndez, F. et al. BULL Database – Spanish Basin attributes for Unravelling Learning in Large-sample hydrology. Sci Data 11, 737 (2024). https://doi.org/10.1038/s41597-024-03594-5

4) L65-75. There are a lot of acronyms in this section. Are they all needed?

**Response:** We believe that these acronyms are necessary for clarity, as each represents a key organization or dataset that plays a distinct role in providing hydrometeorological data for CAMELS-IND. The Central Water Commission (CWC) provides streamflow observations through India-WRIS, the India Meteorological Department (IMD) is responsible for measuring meteorological variables across India, and the National Centre for Medium Range Weather Forecasting (NCMRWF) provides additional meteorological variables via the IMDAA reanalysis data.

5) L90. Why is it 'around 211 catchment attributes'? If you are providing 211 catchment attributes then you don't need the word 'around' in this sentence.

**Response:** Thank you for the suggestion. We have removed word '*around*' in this sentence. [Line 91]

6) L127. What do you mean by 'reliable metadata' – can you be more specific here? What metadata are you considering?

**Response:** Thanks for raising this point. By 'reliable metadata,' we are referring to the comprehensive quality-control checks performed by Goteti (2023) on key aspects of station metadata. Specifically, these checks included verifying the availability and accuracy of station coordinates and descriptions, cross-validating station descriptions with Google Maps or OSM, confirming that reference landmarks are near the station and match station names, assessing the availability and accuracy of CWC's catchment area estimates, and ensuring the adequacy of the delineated catchment and river network.

The revised statement now reads as: "*Given the existing challenges in validating and extracting information from available datasets in India, the GHI has introduced the first quality-controlled metadata in GIS format and listed 472 catchments with consistent and verified metadata out of the 645 gauge stations in Peninsular India (Goteti, 2023). The quality control performed by Goteti (2023) addressed essential metadata aspects, including the accuracy of station coordinates, consistency of station descriptions, and verification of delineated catchment area estimates.*" [Line 127-131]

Goteti, G.: Geospatial dataset for hydrologic analyses in India (GHI): a quality-controlled dataset on river gauges, catchment boundaries and hydrometeorological time series, Earth Syst. Sci. Data, 15, 4389–4415, https://doi.org/10.5194/essd-15-4389-2023, 2023.

7) Figure 1b. It is really hard to see the basin codes on this map – can you make them bigger or a different colour?

**Response:** Thank you for your suggestion. We have updated Figure 1b to improve the visibility of the basin codes by adding a white background to the annotations.

**8) L146.** I would create a new section here called 'Hydrological timeseries' or something similar.Or I would rename section 3 to make it clear to readers that the description of the hydrological timeseries is located here.

**Response:** Thank you for your suggestion. We have renamed Section 3 to "Catchments and Hydrological Time Series" and organized the content into two subsections: 3.1 Catchment Description and 3.2 Hydrological Time Series. [Line 122, 123, 148]

**9) L146.** Can you add some more detail on how the river flow data are compiled for these catchments? Do they undergo any quality assurance or quality control checks before they are published online? Are there flags on any suspect data? Did you perform any checks on the flow data (i.e. for negative values, outliers, multiple consecutive values).

**Response:** Thank you for your comment. We have included details on data compilation and quality control performed by India-WRIS. India-WRIS is a centralized platform that brings together all the publicly available data and information related to water resources in India. "*To ensure data quality, the dataset on India-WRIS undergoes primary validation through the Surface Water Data Entry System (SWDES) and subsequent processing for standardization. The standard operating procedures and data processing protocols are comprehensively described in various user manuals (Lohani, 2012).*" [Line 150-152]. In addition, "*We performed primary validation of the flow data for negative values and outliers, and no such anomalies were observed in the dataset.*" [Line 158-159]

Lohani, A. K.: Surface Water Data Processing Using SWDES, National Institute of Hydrology, 2012. http://117.252.14.250:8080/jspui/handle/123456789/5454

**10) L153-154.** You need to be much clearer here of the data availability of observed streamflow for the catchments. I would argue that 'most' stations do not have reliable data availability. Why do so many of the stations have no streamflow data?

**Response:** Thanks for raising this concern. We have revised the sentence and made changes in Figure 2b to improve clarity as follows: "*our preliminary analysis shows that 228 catchments have streamflow data availability for over 30% of the period between 1980 to 2020 (Fig. 2b).*

*In addition to these 228 catchments, there are 60 catchments with partial streamflow records, ranging from 2.4% (approximately one year) to 30% (about 12 years) of the period. Although these may not meet the threshold for extensive hydrological analysis, they offer valuable opportunities for pseudo-ungauged studies and model testing or validation in data-limited environments.*" [Line 159-163, and Figure 2b].

As stated earlier, the observed streamflow data was compiled from the India-WRIS portal, which was launched in 2019. Since its launch, continuous efforts have been made to digitize the available data and update the information on the portal. We anticipate that, with time, observations from the rest of the gauges will be made available for users, and we will include this data in CAMELS-IND as and when it becomes accessible.

11) L165. This needs more detail on how the gridded rainfall and temperature datasets were produced- is it from observed data (i.e. from rain gauges or weather stations) that are then interpolated on a grid, or from reanalysis data? After reading further in the text, I realise that a lot of this information is in Section 7 but it needs to come earlier in the paper.

**Response:** Thank you for your suggestion. We have moved the details of the data sources for the meteorological time series to Section 4 in the revised manuscript. [Line 178-182, 185-188]

12) Figure 4. It would be helpful to add the map here of the mean precipitation (Fig A3) into this plot to give readers an understanding of how much rainfall falls on average across these catchments.

**Response:** Thank you for your suggestion. We have included mean annual precipitation in Figure 4. [Line 275, Figure 4a]

13) L250. Please quantify and provide numbers for 'Higher mean daily precipitation', 'precipitation decreases', 'moderate precipitation', 'moderate magnitudes are in the central and eastern', 'high values'.

**Response:** We have revised the sentence and provided numbers as follows: "*Higher mean annual precipitation (> 2500 mm) is observed in the Western Ghats region, and the precipitation decreases (< 1000 mm) towards the central part of the region (Fig. 4a). The*

*northern and eastern parts of the region exhibit moderate precipitation in the range of 1500 to 2500 mm.*" [Line 251-253]

"*The spatial patterns of PET indicate moderate values of 1000 to 1250 mm in the central and northern parts, with higher values exceeding 1250 mm in the lower Western Ghats. AET shows a similar trend, with values below 750 mm in the central and northern regions and over 1250 mm in the lower Western Ghats (Fig. A3)*" [Line 272-274]

14) L262. How many gauges did you calculate hydrological indices for?

**Response:** Thank you for raising this question. We have clarified that "*The hydrological signatures were computed for 228 catchments with streamflow data available for at least 30% of the period between 1980 and 2020.*" [Line 282-283]

"*Additionally, due to seasonal precipitation patterns in India, we also computed seasonal flow and its variability, providing quartiles of flow for the southwest monsoon season. For this purpose, we also included gauges with available streamflow observations during specific seasons with less than 20% missing values for all months.*" [Line 284-286]

15) L296. Out of interest, what causes the high runoff ratios along the southwest coast?

**Response:** Thank you for your question. The high runoff ratios along the southwest coast are primarily due to the region's steep topography, with catchment slopes exceeding 16% and drainage path slopes over 20 m/km (**Fig. 3 c-d**). Additionally, these catchments are relatively small. These characteristics contribute to rapid runoff and reduced infiltration, resulting in higher runoff ratios compared to other regions.

16) L395. How do you define a large and medium dam?

**Response:** Thank you for your question. We compiled the dam data from the Central Water Commission (CWC) available through India-WRIS, which defines large dams based on the criteria established by the International Commission on Large Dams (ICOLD). According to ICOLD, a large dam is one that is over 15 meters in height from its foundation, or between 10 and 15 meters if it meets specific structural or operational conditions (such as having a reservoir

capacity exceeding 1 million cubic meters). For medium dams, CWC generally defines these as dams between 10 and 15 meters in height that do not meet the additional criteria required for large dams. In the manuscript we clarified as follows: "*The spatial distribution of large and medium dams (with height > 10 m) across catchments…*" [Line 415]

**17) L403. Where do we see the reservoir use?**

**Response:** The spatial maps of reservoir use for different purposes have been added in Figure A3. [Line 424, 560]

**18) I would encourage the authors to add CAMELS-IND to the Caravan dataset to aid efforts in global catchment datasets.**

**Response:** Thank you for your suggestion. We plan to add CAMELS-IND to the Caravan dataset, and it will be linked to the CAMELS-IND data repository in the near future to support global catchment dataset efforts.

We believe these changes address the concerns raised and improve the quality and clarity of the manuscript. Thank you once again.

Best regards,

Ashutosh Sharma (on behalf of all co-authors)